# Learning from Label Proportions:
# A Mutual Contamination Framework

**Clayton Scott and Jianxin Zhang**
Electrical Engineering and Computer Science
University of Michigan
Ann Arbor, MI 48109
`{clayscot,jianxinz}@umich.edu`

## Abstract

Learning from label proportions (LLP) is a weakly supervised setting for classification in which unlabeled training instances are grouped into bags, and each bag is annotated with the proportion of each class occurring in that bag. Prior work on LLP has yet to establish a consistent learning procedure, nor does there exist a theoretically justified, general purpose training criterion. In this work we address these two issues by posing LLP in terms of mutual contamination models (MCMs), which have recently been applied successfully to study various other weak supervision settings. In the process, we establish several novel technical results for MCMs, including unbiased losses and generalization error bounds under non-iid sampling plans. We also point out the limitations of a common experimental setting for LLP, and propose a new one based on our MCM framework.

## 1  Introduction

Learning from label proportions (LLP) is a weak supervision setting for classification in which training data come in the form of bags. Each bag contains unlabeled instances and is annotated with the proportion of instances arising from each class. Various methods for LLP have been developed, including those based on support vector machines and related models [28, 39, 38, 26, 8, 18, 32], Bayesian and graphical models [17, 14, 35, 25, 15], deep learning [19, 1, 12, 20, 36], clustering [6, 34], and random forests [33]. In addition, LLP has found various applications including image and video analysis [7, 18], high energy physics [9], vote prediction [35], remote sensing [19, 10], medical image analysis [4], activity recognition [25], and reproductive medicine [15].

Despite the emergence of LLP as a prominent weak learning paradigm, the theoretical underpinnings of LLP have been slow to develop. In particular, prior work has not established an algorithm for LLP that is consistent with respect to a classification performance measure. Furthermore, there does not even exist a general-purpose, theoretically grounded empirical objective for training LLP classifiers.

We propose a statistical framework for LLP based on mutual contamination models (MCMs), which have been used previously as models for classification with noisy labels and other weak supervision problems [30, 2, 21, 3, 16]. We use this framework to motivate a principled empirical objective for LLP, prove generalization error bounds associated to two bag generation models, and establish universal consistency with respect to the balanced error rate (BER). The MCM framework further motivates a novel experimental setting that overcomes a limitation of earlier experimental comparisons.

**Related Work.** Quadrianto et al. [27] study an exponential family model for labels given features, and show that the model is characterized by a certain "mean map" parameter that can be estimated in the LLP setting. They also provide Rademacher complexity bounds for the mean map and the associated log-posterior, but do not address a classification performance measure. Patrini et al. [24] extend the work of [27] in several ways, including a generalization error bound on the risk of a

classifier. This bound is expressed in terms of an empirical LLP risk, a "bag Rademacher complexity," and a "label proportion complexity." The authors state that when bags are pure (LPs close to 0 or 1), the last of these terms is small, while for impure bags, the second term is small and the first term increases. While this bound motivates their algorithms, it is not clear how such a bound would imply consistency. Yu et al. [40] study the idea of minimizing the "empirical proportion risk" (EPR), which seeks a classifier that best reproduces the observed LPs. They develop a PAC-style bound on the accuracy of the resulting classifier under the assumption that all bags are very pure. Our work is the first to develop generalization error analysis and universal consistency for a classification performance measure, and we do so under a broadly applicable statistical model on bags.

The literature on LLP has so far yielded two general purpose training objectives that are usable across a variety of learning models. The first of these, the aforementioned EPR, minimizes the average discrepancy between observed and predicted LPs, where discrepancy is often measured by absolute or squared error in the binary case [40, 36, 9], and cross-entropy in the multiclass case [36, 12, 20, 4]. While [40] has been cited as theoretical support for this objective, that paper assumes the bags are very pure, and even provides examples of EPR minimization failure when bags are not sufficiently pure. We offer our own counterexample in the supplemental. The second is the combinatorial objective introduced by [39] that incorporates the unknown labels as variables in the optimization, and jointly optimizes a conventional classification empirical risk together with a term (usually EPR) that encourages correctness of the imputed labels [39, 38, 19, 26, 8, 32, 33, 18, 12]. To our knowledge there is also no statistical theory supporting this objective. In contrast, we propose a theoretically grounded, general purpose criterion for training LLP models.

We also mention that the vast majority of LLP methodology papers simulate data for LLP by taking a classification data set, randomly shuffling the data, and sectioning off the data into bags of a given size. This implies that the expected label proportions for all bags are the same, and as bag size increases, all label proportions converge to the class prior probabilities. The case where all LPs are the same is precisely the setting where LLP becomes intractable, and hence these papers report decreasing performance with increasing bag size. In our experiments we propose an alternate sampling scheme that avoids this issue.

Finally, we mention that in an earlier version of this work [31], we addressed LLP not through the lens of MCMs, but using the label noise model of Natarajan et al. [22]. While that approach leads to the same algorithm as shown in Alg. 1 below, the MCM framework is much more natural for LLP, and the results in the present paper have also been extended considerably, e.g., to accommodate within-bag instance dependencies.

**Notation.** Let $\mathcal{X}$ denote the feature space and $\{-1, 1\}$ the label space. For convenience we often abbreviate $-1$ and $+1$ by "-" and "+", and write $\{\pm\} = \{-, +\}$. A binary classification loss function, or *loss* for short, is a function $\ell : \mathbb{R} \times \{-1, 1\} \to \mathbb{R}$ (we allow losses to take negative values). For $\sigma \in \{\pm\}$, denote $\ell_\sigma(t) := \ell(t, \sigma)$. A loss $\ell$ is *Lipschitz (continuous)* if there exists $L$ such that for every $\sigma \in \{\pm\}$, and every $t, t' \in \mathbb{R}$, $|\ell_\sigma(t) - \ell_\sigma(t')| \le L|t - t'|$. The smallest such $L$ for which this property holds is denoted $|\ell|$. Additionally, we define $|\ell|_0 := \max(|\ell_+(0)|, |\ell_-(0)|)$.

A decision function is a measurable function $f : \mathcal{X} \to \mathbb{R}$. The classifier induced by a decision function $f$ is the function $x \mapsto \text{sign}(f(x))$. We will only consider classifiers induced by a decision function. In addition, we will often refer to a decision function as a classifier, in which case we mean the induced classifier. Let $P_+$ and $P_-$ be the class-conditional distributions of the feature vector $X$, and denote $P = (P_-, P_+)$. The performance measure considered in this work is the *balanced error rate* (BER) which, for a given loss $\ell$, and class conditional distributions $P = (P_+, P_-)$, is defined by $\mathcal{E}_P^\ell(f) := \frac{1}{2}\mathbb{E}_{X \sim P_+}[\ell_+(f(X))] + \frac{1}{2}\mathbb{E}_{X \sim P_-}[\ell_-(f(X))]$. The BER is defined without reference to a distribution of the label $Y$, and hence is invariant to changes in this distribution. The BER is a natural frequentist counterpart to the misclassification rate, which associates a prior probability to each class.

For an integer $n$, denote $[n] := \{1, 2, \ldots, n\}$. Given a sequence of numbers $(a_i)_{i \in [m]}$, denote the arithmetic and harmonic means by $\text{AM}(a_i) := \frac{1}{m}\sum_{i \in [m]} a_i$ and $\text{HM}(a_i) := (\frac{1}{m}\sum_{i \in [m]} a_i^{-1})^{-1}$. Finally, define the probability simplex $\Delta^N := \{w \in \mathbb{R}^N \,|\, w_i \ge 0 \,\forall i, \text{ and } \sum_i w_i = 1\}$.

## 2   Mutual Contamination Models

In this section we define MCMs and present new technical results for learning from MCMs that motivate our study of LLP in the next section, and which may also be of independent interest. We will consider collections of instances $X_1, \ldots, X_m \sim \gamma P_+ + (1 - \gamma)P_-$, where $\gamma \in [0, 1]$ and $m$ are fixed. Foreshadowing LLP, we refer to such collections of instances as *bags*.

We adopt the following assumption on bag data generation, which has two options for modeling within-bag dependencies. Suppose there are $L$ total bags with sizes $n_i$, $i \in [L]$, proportions $\gamma_i \in [0, 1]$, and elements $X_{ij}$, $i \in [L], j \in [n_i]$. We assume

The distributions $P_+$ and $P_-$ are the same for all bags. $\gamma_i$ and $m_i$ may vary from bag to bag. If $i \neq r$, then $X_{ij}$ and $X_{rs}$ are independent $\forall j, s$. Furthermore, for all $i$,

**(IIM)** In the *independent instance model*, $X_{ij} \overset{iid}{\sim} \gamma_i P_+ + (1 - \gamma_i)P_-$;
**(IBM)** In the *independent bag model*, the marginal distribution of $X_{ij}$ is $\gamma_i P_+ + (1 - \gamma_i)P_-$.

**(IBM)** allows the instances within each bag to be dependent. Furthermore, any dependence structure, such as a covariance matrix, *may change from bag to bag*. **(IIM)** is a special case of **(IBM)** that allows us to quantify the impact of bag size $n_i$ on generalization error.

### 2.1   Mutual Contamination Models and Unbiased Losses

Recall that $P$ denotes the pair $(P_+, P_-)$. Let $\kappa = (\kappa^+, \kappa^-)$ be such that $\kappa^+ + \kappa^- < 1$. A *mutual contamination model* is the pair $P^\kappa := (P_+^\kappa, P_-^\kappa)$ where

$$P_+^\kappa := (1 - \kappa^+)P_+ + \kappa^+ P_- \qquad \text{and} \qquad P_-^\kappa := (1 - \kappa^-)P_- + \kappa^- P_+.$$

$P_+^\kappa$ and $P_-^\kappa$ may be thought of as noisy or contaminated versions of $P_+$ and $P_-$, respectively, where the contamination arises from the other distribution. MCMs are common models for label noise [30, 21, 3], where $\kappa^\sigma$ may be interpreted as the label noise rates $\mathbb{P}(Y = -\sigma | \tilde{Y} = \sigma)$, where $Y$ and $\tilde{Y}$ are the true and observed labels.

Let $\ell$ denote any loss for binary classification, such as the logistic loss. Given $\ell$ and $\kappa$ define a new loss $\ell^\kappa$ by

$$\ell_\sigma^\kappa(t) := \frac{1 - \kappa^{-\sigma}}{1 - \kappa^- - \kappa^+}\ell_\sigma(t) - \frac{\kappa^{-\sigma}}{1 - \kappa^- - \kappa^+}\ell_{-\sigma}(t), \qquad \sigma \in \{\pm\}.$$

This loss undoes the bias present in the mutual contamination model.

**Proposition 1.** *Consider any $P = (P_+, P_-)$, $\kappa = (\kappa^+, \kappa^-)$ with $\kappa^+ + \kappa^- < 1$, and loss $\ell$. For any $f$ such that all four of the quantities $\mathbb{E}_{X \sim P_\pm}\ell_\pm(f(X))$ exist and are finite, $\mathcal{E}_P^\ell(f) = \mathcal{E}_{P^\kappa}^{\ell^\kappa}(f)$.*

This result mirrors a similar result established by Natarajan et al. [22] under a label-flipping model for label noise, which is the other prominent model for random label noise besides the MCM. The proof simply matches coefficients of $\mathbb{E}_{X \sim P_\pm}\ell_\pm(f(X))$ on either side of the desired identity.

In the supplemental we offer a sufficient condition for $\ell^\kappa$ to be convex. We also show (as an aside) that Prop. 1 enables a simple proof of a known result concerning symmetric losses, i.e., losses for which $\ell(t, 1) + \ell(t, -1)$ is constant, such as the sigmoid loss. In particular, symmetric losses are immune to label noise under MCMs, meaning the original loss $\ell$ can be minimized on data drawn from the MCM and still optimize the clean BER [21, 37, 5].

The significance of Prop. 1 is that $\mathcal{E}_P^\ell(f)$ is the quantity we want to minimize, while $\mathcal{E}_{P^\kappa}^{\ell^\kappa}(f)$ can be estimated given data from an MCM. In particular, given bags $X_1^+, \ldots, X_{n^+}^+ \sim P_+^\kappa$ and $X_1^-, \ldots, X_{n^-}^- \sim P_-^\kappa$, Prop. 1 motivates minimizing the estimate of BER given by

$$\widehat{\mathcal{E}}(f) := \frac{1}{2n^+}\sum_{j=1}^{n^+}\ell_+^\kappa(f(X_j^+)) + \frac{1}{2n^-}\sum_{j=1}^{n^-}\ell_-^\kappa(f(X_j^-)) = \frac{1}{2}\sum_{\sigma \in \{\pm\}}\frac{1}{n^\sigma}\sum_{j=1}^{n^\sigma}\ell_\sigma^\kappa(f(X_j^\sigma))$$

over $f \in \mathcal{F}$, where $\mathcal{F}$ is some class of decision functions. Under our assumptions on the data, this estimator is unbiased.

**Proposition 2.** *Under **(IBM)**, for any $f$ such that the quantities $\mathbb{E}_{X \sim P_\pm}\ell_\pm(f(X))$ exist and are finite, $\mathbb{E}[\widehat{\mathcal{E}}(f)] = \mathcal{E}_P^\ell(f)$.*

## 2.2 Learning from Multiple Mutual Contamination Models

In the next section we view LLP in terms of a more general problem that we now define. Suppose we are given $N$ different MCMs. Each has the same true class-conditional distributions $P_+$ and $P_-$, but possibly different contamination proportions $\kappa_i = (\kappa_i^+, \kappa_i^-)$, $i \in [N]$. Let $P^{\kappa_i} = (P_+^{\kappa_i}, P_-^{\kappa_i})$ denote the $i$th MCM, and assume $\kappa_i^+ + \kappa_i^- < 1$. Now suppose that for each $i \in [N]$, we observe

$$X_{i1}^+, \ldots, X_{in_i^+}^+ \sim P_+^{\kappa_i} := (1 - \kappa_i^+)P_+ + \kappa_i^+ P_-,$$

$$X_{i1}^-, \ldots, X_{in_i^-}^- \sim P_-^{\kappa_i} := (1 - \kappa_i^-)P_- + \kappa_i^- P_+.$$

The problem of *learning from multiple mutual contamination models* (LMMCM) is to use all of the above data to design a single classifier that minimizes the clean BER $\mathcal{E}_P^\ell$.

A natural approach to this problem is to minimize the weighted empirical risk

$$\widehat{\mathcal{E}}_w(f) := \sum_{i=1}^N w_i \widehat{\mathcal{E}}_i(f), \quad \text{where} \quad \widehat{\mathcal{E}}_i(f) := \frac{1}{2n_i^+} \sum_{j=1}^{n_i^+} \ell_+^{\kappa_i}(f(X_{ij}^+)) + \frac{1}{2n_i^-} \sum_{j=1}^{n_i^-} \ell_-^{\kappa_i}(f(X_{ij}^-))$$

and $w \in \Delta^N$. By Prop. 1, under **(IBM)** each $\widehat{\mathcal{E}}_i(f)$ is an unbiased estimate of $\mathcal{E}_P^\ell(f)$, and therefore so is $\widehat{\mathcal{E}}_w(f)$. This leads to the question whether we may set $w$ to minimize some notion of "variance." Intuitively, MCMs $P^{\kappa_i}$ with less corruption should receive larger weights. We confirm this intuition by quantifying "variance" in terms of generalization error, and choosing $w_i$ to optimize a generalization error bound (GEB). Our GEBs uses two weighted, multi-sample extensions of Rademacher complexity, corresponding to **(IIM)** and **(IBM)**, that we now introduce.

Let $S$ denote all the data $X_{ij}^\sigma$ from $N$ MCMs as described above.

**Definition 3.** *Let $\mathcal{F}$ be a class of decision functions. Assume that $\sup_{f \in \mathcal{F}} \sup_{x \in \mathcal{X}} |f(x)| < \infty$. For any $c \in \mathbb{R}_{\geq 0}^N$, define*

$$\mathfrak{R}_c^I(\mathcal{F}) := \mathbb{E}_S \mathbb{E}_{(\epsilon_{ij}^\sigma)} \left[ \sup_{f \in \mathcal{F}} \sum_{i=1}^N c_i \sum_{\sigma \in \{\pm\}} \frac{1}{2n_i^\sigma} \sum_{j=1}^{n_i^\sigma} \epsilon_{ij}^\sigma f(X_{ij}^\sigma) \right], \tag{1}$$

*and*

$$\mathfrak{R}_c^B(\mathcal{F}) := \mathbb{E}_S \mathbb{E}_{((\sigma_i, X_i) \sim \widehat{P}^{\kappa_i})_{i \in [N]}} \mathbb{E}_{(\epsilon_i)} \left[ \sup_{f \in \mathcal{F}} \sum_{i=1}^N \epsilon_i c_i f(X_i) \right], \tag{2}$$

*where $\epsilon_{ij}^\sigma, \epsilon_i \overset{iid}{\sim} \mathrm{unif}(\{-1, 1\})$ are Rademacher random variables and $\widehat{P}^{\kappa_i}$ is the distribution that selects $\sigma_i \sim \mathrm{unif}(\{-1, 1\})$, and then draws $X_i$ uniformly from $X_{i,1}^{\sigma_i}, \ldots, X_{i,n_i^{\sigma_i}}^{\sigma_i}$.*

The inner two summations in (1) reflect an adaptation of the usual Rademacher complexity to the BER, and the outer summation reflects the multiple MCMs. Eqn. (2) may be seen as a modification of (1) where the inner two sums are viewed as an empirical expectation that is pulled out of the supremum. If $\mathcal{F}$ satisfies the following, then $\mathfrak{R}_c^I(\mathcal{F})$ and $\mathfrak{R}_c^B(\mathcal{F})$ are bounded by tractable expressions.

**(SR)** There exist constants $A$ and $B$ such that $\sup_{f \in \mathcal{F}} \sup_{x \in \mathcal{X}} |f(x)| \leq A$, and for all $M$, $x_1, \ldots, x_M \in \mathcal{X}$, and $a \in \mathbb{R}_{\geq 0}^M$,

$$\mathbb{E}_{(\epsilon_i)} \left[ \sup_{f \in \mathcal{F}} \sum_{i=1}^M \epsilon_i a_i f(x_i) \right] \leq B \sqrt{\sum_{i=1}^M a_i^2}.$$

As one example of an $\mathcal{F}$ satisfying **(SR)**, let $k$ be a symmetric positive definite (SPD) kernel, bounded[1] by $K$, and let $\mathcal{H}$ be the associated reproducing kernel Hilbert space (RKHS). Let $\mathcal{F}_{K,R}^k$ denote the ball of radius $R$, centered at 0, in $\mathcal{H}$. As a second example, assume $\mathcal{X} \subset \mathbb{R}^d$ and

$\|\mathcal{X}\|_2 := \sup_{x \in \mathcal{X}} \|x\|_2 < \infty$, where $\|\cdot\|_2$ is the Euclidean norm. Let $\alpha, \beta \in \mathbb{R}_+^M$ and denote $[x]_+ = \max(0, x)$. Define the class of two-layer neural networks with ReLU activation by

$$\mathcal{F}_{\alpha,\beta}^{\mathrm{NN}} = \{f(x) = v^T [Ux]_+ : v \in \mathbb{R}^h, U \in \mathbb{R}^{h \times d}, |v_i| \le \alpha_i, \|u_i\|_2 \le \beta_i, i = 1, 2, \ldots, h\}.$$

**Proposition 4.** $\mathcal{F}_{K,R}^k$ and $\mathcal{F}_{\alpha,\beta}^{NN}$ satisfy **(SR)** with $(A, B) = (RK, RK)$ and $(A, B) = (\|\alpha\|_2 \|\beta\|_2 \|\mathcal{X}\|_2, 2\langle \alpha, \beta \rangle \|\mathcal{X}\|_2)$, respectively.

We emphasize that other classes $\mathcal{F}$ admit quantitative bounds on $\mathfrak{R}_c^I(\mathcal{F})$ and $\mathfrak{R}_c^B(\mathcal{F})$ that do not necessarily conform to **(SR)** , and that can also be leveraged as we do below. We focus on **(SR)** because the GEBs simplify considerably, making it possible to derive closed form expressions for the optimal $w_i$. Below we write $\overset{(\mathbf{SR})}{\le}$ to indicate an upper bound that holds provided **(SR)** is true.

Our first main result establishes GEBs for LMMCM under both **(IIM)** and **(IBM)** .

**Theorem 5.** *Let $S$ collect all the data $(X_{ij}^\sigma)$ from $N$ MCMs with common base distributions $P_+, P_-$, and contamination proportions $\kappa_i = (\kappa_i^+, \kappa_i^-)$ satisfying $\kappa_i^- + \kappa_i^+ < 1$. Let $\mathcal{F}$ be a class of decision functions such that $A = \sup_{f \in \mathcal{F}} \sup_{x \in \mathcal{X}} |f(x)| < \infty$, $\ell$ a Lipschitz loss, $w \in \Delta^N$, and $\delta > 0$. Under **(IIM)** , with probability $\ge 1 - \delta$ wrt the draw of $S$,*

$$\sup_{f \in \mathcal{F}} \left| \widehat{\mathcal{E}}_w(f) - \mathcal{E}(f) \right| \le 2\mathfrak{R}_c^I(\mathcal{F}) + C \sqrt{\sum_{i=1}^N \frac{w_i^2}{\bar{n}_i (1 - \kappa_i^- - \kappa_i^+)^2}} \overset{(\mathbf{SR})}{\le} D \sqrt{\sum_{i=1}^N \frac{w_i^2}{\bar{n}_i (1 - \kappa_i^- - \kappa_i^+)^2}}$$

(3)

*where $\bar{n}_i := \mathrm{HM}(n_i^-, n_i^+)$, $c_i = w_i |\ell| / (1 - \kappa_i^- - \kappa_i^+)$, $C = (1 + A|\ell|)\sqrt{\log(2/\delta)}$, and $D = 2B|\ell| + C$. Under **(IBM)** , the same statement holds after replacing $\mathfrak{R}_c^I(\mathcal{F}) \to \mathfrak{R}_c^B(\mathcal{F})$ and $\bar{n}_i \to 1$.*

Several remarks are in order. Under **(IIM)** , even in the special case $N = 1$ without noise ($\kappa_1^- = \kappa_1^+ = 0$) the result appears new, and amounts to an adaptation of the standard Rademacher complexity bound to BER. The case $N = 1$ *with* noise can be used to prove consistency (with $\bar{n}_1 \to \infty$) of a discrimination rule for a single $MCM$ given knowledge of, or consistent estimates of $\kappa_1^-, \kappa_1^+$. Previous results of this type have analyzed MCMs via label-flipping models which are less natural [3].

Because the result holds for any $w \in \Delta^N$, as long as the $\kappa_i$ are known a priori, we may set $w$ to optimize the rightmost expressions in (3). This leads to optimal weights $w_i \propto \bar{n}_i (1 - \kappa_i^- - \kappa_i^+)^2$ under **(IIM)** (here and below, replace $\bar{n}_i$ by 1 for **(IBM)** ), which supports our claim that MCMs with more information (larger samples, less noise) should receive more weight. With this choice of weights, the summation in the bound reduces to $\frac{1}{N} \mathrm{HM}(1/\bar{n}_i (1 - \kappa_i^- - \kappa_i^+)^2)$. In contrast, with uniform weights $w_i = 1/N$ the summation equals $\frac{1}{N} \mathrm{AM}(1/\bar{n}_i (1 - \kappa_i^- - \kappa_i^+)^2)$. The harmonic mean is much less sensitive to the presence of outliers, i.e., very noisy MCMs, than the arithmetic. As an illustration, suppose $N = 10$, $n_i = n = 100$, and for $i < N$, $\rho_i^+ = \rho_i^- = 0.01$, while $\rho_N^+ = \rho_N^- = 0.49$. Then the ratio of arithmetic mean to harmonic mean exceeds 100.

## 3 Learning from Label Proportions

In learning from label proportions with binary labels, the learner has access to $(b_1, \widehat{\gamma}_1), \ldots, (b_L, \widehat{\gamma}_L)$, where each $b_i$ is a bag of $n_i$ unlabeled instances, and each $\widehat{\gamma}_i \in [0, 1]$ is the proportion of instances from class 1 in the bag. The goal is to learn an accurate classifier as measured by some performance measure, which in our case we take to be the BER. This choice is already a departure from prior work on LLP, which typically looks at misclassification rate (MCR). The BER is defined without reference to a distribution of the label $Y$, and is thus invariant to changes in this distribution. In other words, BER is immune to shifts in class prevalence, and hence to shifts in the distribution of label proportions.

We adopt the following data generation model for bags. Each bag has a *true label proportion* $\gamma_i \in [0, 1]$. For each $i$, let $(X_{ij}, Y_{ij})$, $j \in [n_i]$, be random variables. The $i$th bag is formed from $(X_{ij})_{j \in [n_i]}$, and the *observed* or *empirical label proportion* is $\widehat{\gamma}_i = \frac{1}{n_i} \sum_j \frac{Y_{ij} + 1}{2}$. Let $\boldsymbol{\gamma}, \boldsymbol{Y}$, and $\boldsymbol{X}$ be vectors collecting all of the values of $\gamma_i, Y_{ij}$, and $X_{ij}$, respectively. We assume

The distributions $P_+$ and $P_-$ are the same for all bags. The $\gamma_i$ may be random, and the sizes $n_i$ are nonrandom. Conditioned on $\boldsymbol{\gamma}$, if $i \neq r$, then $X_{ij}$ and $X_{rs}$ are independent $\forall j, s$. Furthermore, conditioned on $\boldsymbol{\gamma}$, for bag $i$

**(CIIM)** In the *conditionally independent instance model*, $\frac{Y_{ij}+1}{2} \stackrel{iid}{\sim}$ Bernoulli$(\gamma_i)$ and conditioned on $Y_{i1}, \ldots, Y_{in_i}$, $X_{i1}, \ldots, X_{in_i}$ are independent with $X_{ij} \sim P_{Y_{ij}}$.

**(CIBM)** In the *conditionally independent bag model*, $\mathbb{E}[\widehat{\gamma}_i] = \gamma_i$ and for each $j$, the distribution of $X_{ij}|Y_{i1}, \ldots, Y_{in_i}$ is $P_{Y_{ij}}$.

Under **(CIBM)**, conditioned on $\boldsymbol{\gamma}$, for bag $i$ the labels $Y_{i1}, \ldots, Y_{in_i}$ may be dependent, and given these labels the instances $X_{ij}$ may also be dependent. Furthermore, the dependence structure may change from bag to bag. This means that given its label, the distribution of an instance is still dependent on its bag, in contrast to prior work [27]. We also allow that the $\gamma_i$ may be dependent, so that without conditioning on $\boldsymbol{\gamma}$, the bags themselves may be dependent.

As in the previous section, the significance of our model is that it provides for (conditionally) unbiased estimates of BER as we describe below. Indeed, if we view $\boldsymbol{\gamma}$ as fixed, **(CIIM)** clearly implies **(IIM)** (in fact, the two independent instance models are equivalent). However, it is not the case that **(CIBM)** implies **(IBM)** – the introduction of the latent labels allows for a more general independent bag model while still ensuring unbiased BER estimates. A strengthening of **(CIBM)**, namely

**(CIBM')** For each $j$, $\mathbb{E}[\frac{Y_{ij}+1}{2}] = \gamma_i$ and the distribution of $X_{ij}|Y_{i1}, \ldots, Y_{in_i}$ is $P_{Y_{ij}}$

does imply **(IBM)** (still viewing $\boldsymbol{\gamma}$ as fixed), as we show in the supplemental.

In this section we propose to reduce LLP to the setting of the previous section by pairing the bags, so that each pair of bags constitutes an MCM.

## 3.1 LLP when True Label Proportions are Known

We first consider the less realistic setting where the $\gamma_i$ are deterministic and *known*. In this situation we may reduce LLP to LMMCM by pairing bags. In particular, we re-index the bags and let $(b_i^-, \gamma_i^-)$ and $(b_i^+, \gamma_i^+)$ constitute the $i$th pair of bags, such that $\gamma_i^- < \gamma_i^+$. The bags may be paired in any way that depends on $\gamma_1, \ldots, \gamma_L$, subject to $\gamma_i^- < \gamma_i^+ \ \forall i$. We also assume the total number of bags is $L = 2N$, so that the number of bag pairs is $N$.

If we set $\kappa_i = (\kappa_i^+, \kappa_i^-) := (1 - \gamma_i^+, \gamma_i^-)$, then we are in the setting of LMMCM described in the previous setting. Furthermore, $1 - \kappa_i^- - \kappa_i^+ = \gamma_i^+ - \gamma_i^- > 0$. Therefore we may apply all of the theory developed in the previous section without modification. Since $\boldsymbol{\gamma}$ is deterministic, **(CIIM)** and **(CIBM)'** imply **(IIM)** and **(IBM)** as discussed above, and we may simply apply Theorem 5 to obtain GEBs for LLP. Choosing weights $w_i$ to minimize the **(SR)** form yields final bounds proportional to the square root of $\frac{1}{N} \text{HM}(1/(\bar{n}_i(\gamma_i^+ - \gamma_i^-)^2)) = (\sum_i \bar{n}_i(\gamma_i^+ - \gamma_i^-)^2)^{-1}$ (under **(CIBM')** replace $\bar{n}_i \to 1$). In the LLP setting, we may further optimize this bound by optimizing the pairing of bags. This leads to an integer program known as the "maximum weighted (perfect) matching" problem. An exact algorithm to solve it was given by Edmonds [13], and several approximate algorithms also exist for large scale problems [11]. See supplemental for additional details.

If $\boldsymbol{\gamma}$ is random, and the $\gamma_i$ are distinct (which occurs w. p. 1, e.g., if $\boldsymbol{\gamma}$ is jointly continuous), Theorem 5 still holds conditioned on $\boldsymbol{\gamma}$, and therefore unconditionally by the law of total expectation.

Although the $\gamma_i$ are typically unknown in practice, the above discussion still yields a useful algorithm: simply "plug in" $\widehat{\gamma}_i$ for $\gamma_i$ and proceed to minimize $\widehat{\mathcal{E}}_w(f)$ (with optimally paired bags and optimized weights) over $\mathcal{F}$. A description of the learning procedure, which we use in our experiments, is presented in Algorithm 1.

---
**Algorithm 1** Plug-in approach to LLP via LMMCM (outline)
---
1: **Input:** $(b_1, \widehat{\gamma}_1), \ldots, (b_{2N}, \widehat{\gamma}_{2N})$, model class $\mathcal{F}$, loss $\ell$, tuning parameters
2: **procedure** LLP-LMMCM
3:     Solve weighted matching problem to find pairings maximizing $\sum_i w_i \propto \bar{n}_i (\widehat{\gamma}_i^+ - \widehat{\gamma}_i^-)^2$ (see supplemental)
4:     Set $\kappa_i = (1 - \widehat{\gamma}_i^+, \widehat{\gamma}_i^-)$ and optimal weights $w_i \propto \bar{n}_i (\widehat{\gamma}_i^+ - \widehat{\gamma}_i^-)^2$
5:     Minimize $\widehat{\mathcal{E}}_w(f)$ over $\mathcal{F}$, perhaps with regularization
---

## 3.2 Consistent Learning from Label Proportions

When the true label proportions are not known, as is usually the case in practice, it is difficult to establish consistency of the plug-in approach without restrictive assumptions. This is because the $\widehat{\gamma}_i$ are random, and so there is always some nonnegligible probability that in each pair, the bag with larger $\gamma_i$ will be misidentified. This problem is especially pronounced for very small bag sizes. For example, if two bags with $\gamma_1 = .45$ and $\gamma_2 = .55$ are paired, and the bag sizes are 8 with independent labels, the probability that $\widehat{\gamma}_2 < \widehat{\gamma}_1$ is .26. One approach to overcoming this issue is to have the bag sizes $n_i^\sigma$ tend to $\infty$ asymptotically, in which case $\widehat{\gamma}_i \overset{a.s.}{\to} \gamma_i$. This is a less interesting setting, however, because the learner can discard all but one pair of bags and still achieve consistency using existing techniques for learning in MCMs [3]. Furthermore, the bag size is often fixed in applications.

We propose an approach based on merging the original "small bags" to form "big bags," and then applying the approach of Section 3.1. For convenience assume all original (small) bags have the same size $n_i = n$ moving forward. Let $K$ be an integer and assume $N$ is a multiple of $K$ for convenience, $N = MK$. As before, let $(b_i, \widehat{\gamma}_i)$, $i \in [2N]$, be the original, unpaired bags of size $n$. We refer to a *K-merging scheme* as any procedure that takes the original unpaired bags of size $n$ and combines them, using knowledge of the $\widehat{\gamma}_i$, to form paired bags of size $nK$. Let the paired bags be denoted $(B_i^+, \widehat{\Gamma}_i^+)$ and $(B_i^-, \widehat{\Gamma}_i^-)$, $i \in [M]$. Let $I_i^\sigma$ denote the original indices of the small bags comprising $B_i^\sigma$, so that $B_i^\sigma = \cup_{j \in I_i^+} b_i$ and $\widehat{\Gamma}_i^\sigma = \frac{1}{K} \sum_{j \in I_i^\sigma} \widehat{\gamma}_j^\sigma$.

We offer two examples of $K$-merging schemes. The first, called the *blockwise-pairwise (BP) scheme*, simply takes the original small bags in their given order. The $i$th block of 2K consecutive small bags are used to form the $i$th pair of big bags. This is done by considering consecutive, nonoverlapping pairs of small bags and assigning the small bag with larger $\widehat{\gamma}_i$ to $B_i^+$. Using notation, we define $I_i^+ = \{j \in [2K(i-1) + 1 : 2Ki] \mid j \text{ is odd and } \widehat{\gamma}_j \geq \widehat{\gamma}_{j+1} \text{ or } j \text{ is even and } \widehat{\gamma}_j \geq \widehat{\gamma}_{j-1}\}$ and $I_i^- = [2K(i-1) + 1 : 2Ki] \backslash I_i^+$ (ties may be broken arbitrarily). The *blockwise-max (BM) scheme* is like BP, except that for each block of $2K$ small bags, the $K$ small bags with largest $\widehat{\gamma}_j$ are assigned to the positive bag. One can imagine more elaborate schemes that are not blockwise. We say that scheme 1 *dominates* scheme 2 if, with probability 1, for every $i$, $\widehat{\Gamma}_i^+ - \widehat{\Gamma}_i^-$ for scheme 1 is at least as large as it is for scheme 2. For example, BM dominates BP.

Next, we form the modified weighted empirical risk. For each $i \in [M]$ and $\sigma \in \{\pm\}$, let $(X_{ij}^\sigma)$, $j \in [nK]$, denote the elements of $B_i^\sigma$, and $(Y_{ij}^\sigma)$ the associated labels. Also set $\widehat{\kappa}_i = (1 - \widehat{\Gamma}_i^+, \widehat{\Gamma}_i^-)$. Let $w \in \Delta^M$ such that $w_i \propto (\widehat{\Gamma}_i^+ - \widehat{\Gamma}_i^-)^2$, and define

$$\tilde{\mathcal{E}}(f) := \sum_{i=1}^M w_i \tilde{\mathcal{E}}_i(f) \qquad \text{where} \qquad \tilde{\mathcal{E}}_i(f) := \left[ \frac{1}{2n} \sum_{\sigma \in \{\pm\}} \sum_{j=1}^{nK} \ell_\sigma^{\widehat{\kappa}_i}(f(X_{ij}^\sigma)) \right].$$

In the proof of Thm. 6, we show that under **(CIBM)**, with high probability, $\tilde{\mathcal{E}}_i(f)$ is an unbiased estimate for $\mathcal{E}_P^\ell(f)$ when conditioned on $\gamma$ and $Y$.

To state our main result we adopt the following assumption on the distribution of label proportions.

**(LP)** There exist $\Delta, \tau > 0$ such that the sequence of random variables $Z_j = \mathbf{1}_{\{|\gamma_j - \gamma_{j+1}| < \Delta\}}$ satisfies the following. For every $J \subseteq [2N-1]$, $\mathbb{P}(\prod_{j \in J} Z_j = 1) \leq \tau^{|J|}$.

This condition is satisfied if the $\gamma_i$ are iid draws from any non-constant distribution. However, it also allows for the $\gamma_i$ to be correlated. As one example, let $(w_j)$ be iid random variables with support

$\supseteq [-1, 1]$. **(LP)** is satisfied if $\gamma_{j+1} = \gamma_j + \underline{w}_j$, where $\underline{w}_j$ is the truncation of $w_j$ to $[-\gamma_j, 1 - \gamma_j]$. The point of **(LP)** is that it offers a dependence setting where a one-sided version of Hoeffding's inequality holds, which allows us to conclude that with high probability, for all odd $j \in [2N]$, $|\gamma_j - \gamma_{j+1}| \geq \Delta$ for approximately $N(1 - \tau)$ of the original pairs of small bags [23].

We now state our main result. Define $\Gamma_i^+ = \mathbb{E}_{\boldsymbol{Y}|\boldsymbol{\gamma}}[\widehat{\Gamma}_i^+]$ and $\Gamma_i^- = \mathbb{E}_{\boldsymbol{Y}|\boldsymbol{\gamma}}[\widehat{\Gamma}_i^-]$.

**Theorem 6.** *Let* **(LP)** *hold. Let* $\epsilon_0 \in (0, \Delta(1 - \tau))$. *Let* $\ell$ *be a Lipschitz loss and let* $\mathcal{F}$ *satisfy* $\sup_{x \in \mathcal{X}, f \in \mathcal{F}} |f(x)| \leq A < \infty$. *Let* $\epsilon \in (0, \frac{\Delta(1-\tau)-\epsilon_0}{1+\Delta}]$ *and* $\delta \in (0, 1]$. *For the BP merging scheme, under* **(CIIM)** *, with probability at least* $1 - \delta - 2\frac{N}{K}e^{-2K\epsilon^2}$ *with respect to the draw of* $\boldsymbol{\gamma}, \boldsymbol{Y}, \boldsymbol{X}$,

$$\widehat{\Gamma}_i^+ - \widehat{\Gamma}_i^- \geq \Gamma_i^+ - \Gamma_i^- - \epsilon \geq \epsilon_0$$

*and*

$$\sup_{f \in \mathcal{F}} \left| \tilde{\mathcal{E}}(f) - \mathcal{E}(f) \right| \leq 2\mathfrak{R}_c^I(\mathcal{F}) + C\sqrt{\frac{\text{HM}((\Gamma_i^+ - \Gamma_i^- - \epsilon)^{-2})}{2(N/K)n}} \overset{\textbf{(SR)}}{\leq} D\sqrt{\frac{\text{HM}((\Gamma_i^+ - \Gamma_i^- - \epsilon)^{-2})}{2(N/K)n}},$$

$$\tag{4}$$

*where* $c_i = w_i|\ell|/(\Gamma_i^+ - \Gamma_i^- - \epsilon)$, $C = (1 + A|\ell|)\sqrt{\log(2/\delta)}$, *and* $D = 2B|\ell| + C$. *Under* **(CIBM)**, *the same bounds hold with the same probability if we substitute* $\mathfrak{R}_c^I(\mathcal{F}) \to \mathfrak{R}_c^B(\mathcal{F})$ *and* $n \to 1$.

This result states that BP achieves essentially the same bound (modulo $\epsilon$) as if we applied LMMCM to the big bags with *known* $\Gamma_i^+, \Gamma_i^-$. We also note that there is no restriction on bag size $n$. A corollary of this result also applies to any scheme that dominates BP, as we explain in the supplemental.

Theorem 6 implies a consistent learning algorithm for LLP under both **(CIIM)** and **(CIBM)** , using any merging scheme that dominates BP. To achieve consistency the bound should tend to zero while the confidence tends to 1, as $N \to \infty$. Even with $n$ fixed, this is true provided $K \to \infty$ and $N/K \to \infty$ as $N \to \infty$, such that $N = O(K^\beta)$ for some $\beta > 0$. Beyond that, standard arguments may be applied to arrive at a formal consistency result. In the supplemental we state such a result for completeness. Here the consistency is *universal* in that it makes no assumptions on $P_-$ or $P_+$.

Our consistency result is for BER defined with an arbitrary loss $\ell$. If we desire consistency for BER with 0-1 loss, but still want a tractable algorithm, we can achieve this by taking $\ell$ to be classification calibrated, as discussed in the supplemental.

## 4 Experiments

The vast majority of LLP methodology papers simulate data for LLP by taking a classification data set, randomly shuffling the data, and sectioning off the data into bags of a certain size. This implies that the expected label proportions for all bags are the same, and as bag size increases, all label proportions converge to the class prior probabilities. The case where all LPs are the same is precisely the setting where LLP becomes intractable, and hence these papers report decreasing performance with increasing bag size.

We propose an alternate sampling scheme inspired by our MCM framework.[2] Each experiment is based on a classification data set, a distribution of LPs, and the bag size $n$. For each dataset, the total number of training instances $T$ is fixed, so that the number of bags is $T/n$. We consider the Adult ($T = 8192$) and MAGIC Gamma Ray Telescope ($T = 6144$) datasets (both available from the UCI repository[3]), LPs that are iid uniform on $[0, \frac{1}{2}]$ and on $[\frac{1}{2}, 1]$, and bag sizes $n \in \{8, 32, 128, 512\}$. The total number of experimental settings is thus $2 \times 2 \times 4 = 16$. The numerical features in both datasets are standardized to have 0 mean and unit variance, the categorical features are one-hot encoded.

We implement a method based on our general approach (see Algorithm 1) by taking $\ell$ to be the logistic loss, $\mathcal{F}$ to be the RKHS associated to a Gaussian kernel $k$, and selecting $f \in \mathcal{F}$ by minimizing $\widehat{\mathcal{E}}_w(f) + \lambda\|f\|_{\mathcal{F}}^2$. By the representer theorem [29], the minimizer of this objective has the form $f(x) = \sum_i \alpha_i k(x, x_i)$ where $\alpha_i \in \mathbb{R}$ and $x_i$ ranges over all training instances. Our Python implementation uses SciPy's L-BFGS routine to find the optimal $\alpha_i$. The kernel parameter is

computed by $\frac{1}{d*Var(X)}$ where $d$ is the number of features and $Var(X)$ is the variance of the data matrix, and the parameter $\lambda \in \{1, 10^{-1}, 10^{-2}, \ldots, 10^{-5}\}$ is chosen by 5-fold cross validation. We tried the EPR as a criterion for model selection but found our own criterion to be better. For each dataset, our implementation runs all 8 settings in roughly 50 minutes using 48 cores.

We compare against InvCal [28] and alter-$\propto$SVM [39], the two most common reference methods in LLP, using Matlab implementations provided by the authors of [39]. Those methods are designed to optimize accuracy, whereas ours is designed to optimize BER. For a fair comparison, we employed a third criterion. In particular, for each method we shift the decision function's threshold to generate an ROC curve and evaluate the area under the curve (AUC) using all data that was not used for training. For each experimental setting, the reported AUC and standard deviation reflect the average results over 5 randomized trials. Additional experimental details are found in the supplement.

The results are reported in Table 1. Bold numbers indicate that a method's mean AUC was the largest for that experimental setting. We see that for the smallest bag size, the methods all perform comparably, while for larger bag sizes, LMMCM exhibits far less degradation in performance. Using the Wilcoxon signed-rank test, we find that LMMCM outperforms InvCal and $\propto$SVM with p-value < 0.005.

In the supplement we present a variant of Table 1 where the number of bags $N$ is fixed. This table leads to similar conclusions.

Table 1: AUC. Column header indicates bag size.

| Data set, LP dist | Method | 8 | 32 | 128 | 512 |
|---|---|---|---|---|---|
| Adult, $\left[0, \frac{1}{2}\right]$ | InvCal | $0.8720 \pm 0.0035$ | $0.8672 \pm 0.0067$ | $0.8537 \pm 0.0101$ | $0.7256 \pm 0.0159$ |
| | alter-$\propto$SVM | $0.8586 \pm 0.0185$ | $0.7394 \pm 0.0686$ | $0.7260 \pm 0.0953$ | $0.6876 \pm 0.1219$ |
| | LMMCM | $\mathbf{0.8728 \pm 0.0019}$ | $\mathbf{0.8693 \pm 0.0047}$ | $\mathbf{0.8669 \pm 0.0041}$ | $\mathbf{0.8674 \pm 0.0040}$ |
| Adult, $\left[\frac{1}{2}, 1\right]$ | InvCal | $\mathbf{0.8680 \pm 0.0021}$ | $0.8598 \pm 0.0073$ | $0.8284 \pm 0.0093$ | $0.7480 \pm 0.0500$ |
| | alter-$\propto$SVM | $0.8587 \pm 0.0097$ | $0.7429 \pm 0.1473$ | $0.8204 \pm 0.0318$ | $0.7602 \pm 0.1215$ |
| | LMMCM | $0.8584 \pm 0.0164$ | $\mathbf{0.8644 \pm 0.0052}$ | $\mathbf{0.8601 \pm 0.0045}$ | $\mathbf{0.8500 \pm 0.0186}$ |
| MAGIC, $\left[0, \frac{1}{2}\right]$ | InvCal | $\mathbf{0.8918 \pm 0.0076}$ | $0.8574 \pm 0.0079$ | $0.8295 \pm 0.0139$ | $0.8133 \pm 0.0109$ |
| | alter-$\propto$SVM | $0.8701 \pm 0.0026$ | $0.7704 \pm 0.0818$ | $0.7753 \pm 0.0207$ | $0.6851 \pm 0.1580$ |
| | LMMCM | $0.8909 \pm 0.0077$ | $\mathbf{0.8799 \pm 0.0113}$ | $\mathbf{0.8753 \pm 0.0157}$ | $\mathbf{0.8734 \pm 0.0092}$ |
| MAGIC, $\left[\frac{1}{2}, 1\right]$ | InvCal | $\mathbf{0.8936 \pm 0.0066}$ | $0.8612 \pm 0.0056$ | $0.8180 \pm 0.0092$ | $0.8215 \pm 0.0136$ |
| | alter-$\propto$SVM | $0.8689 \pm 0.0135$ | $0.8219 \pm 0.0218$ | $0.8179 \pm 0.0487$ | $0.7949 \pm 0.0478$ |
| | LMMCM | $0.8911 \pm 0.0083$ | $\mathbf{0.8790 \pm 0.0091}$ | $\mathbf{0.8684 \pm 0.0046}$ | $\mathbf{0.8567 \pm 0.0292}$ |

## 5  Conclusion

We have introduced a principled framework for LLP based on MCMs. We have developed several novel results for MCMs, and used them to develop a statistically consistent procedure and an effective practical algorithm for LLP. The most natural direction for future work is to extend to multiclass.

## Broader Impact

LLP has been discussed as a model for summarizing a fully labeled dataset for public dissemination. The idea is that individual labels are not disclosed, so some degree of privacy is retained. As we show, consistent classification is still possible in this setting. If the two class-conditional distributions are nonoverlapping, labels of training instances can be recovered with no uncertainty by an optimal classifier. If the class-conditional distributions have some overlap, training instances in the nonoverlapping region can still be labeled with no uncertainty, while training instances in the overlapping regions can have their labels guessed with some uncertainty, depending on the degree of overlap.

## Acknowledgments and Disclosure of Funding

The authors thank Laura Balzano, Jeff Fessler, and Mert Pilanci for their input. CS was supported in part by the National Science Foundation under awards 1838179 and 2008074, and by the Department of Defense, Defense Threat Reduction Agency under award HDTRA1-20-2-0002.

## Footnotes

[1] An SPD kernel $k$ is bounded by $K$ if $\sqrt{k(x,x)} \leq K$ for all $x$. For example, the Gaussian kernel $k(x, x') = \exp(-\gamma \|x - x'\|^2)$ is bounded by $K = 1$.

[2]https://github.com/Z-Jianxin/Learning-from-Label-Proportions-A-Mutual-Contamination-Framework

[3]http://archive.ics.uci.edu/ml

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
