[Supplementary Material]

# Supplemental Material for
# Learning from Label Proportions:
# A Mutual Contamination Framework

**Clayton Scott and Jianxin Zhang**
Electrical Engineering and Computer Science
University of Michigan
Ann Arbor, MI 48109
{clayscot,jianxinz}@umich.edu

Section, result, and equation numbers in this supplement are prefixed with an "S", while those from the main document are not.

## S1   Failure Case for Empirical Proportion Risk Minimization

We offer a simple example where minimizing the empirical proportion risk leads to suboptimal performance. Let $P_-$ be uniform on $[0, 1]$, with density $p_-(x) = \mathbf{1}_{\{x \in [0,1]\}}$, and let $P_+$ have the triangular density function $p_+(x) = 2x\mathbf{1}_{\{x \in [0,1]\}}$. Suppose there is a single bag, and that the label proportion is $\gamma = \frac{1}{2}$. Also suppose $\mathcal{F}$ consists of threshold classifiers $f_t(x) = \text{sign}(x - t), t \in [0, 1]$. This class contains the optimal BER classifier (define wrt 0-1 loss) corresponding to $t^* = \frac{1}{2}$. Now suppose we are in the infinite bag-size limit (which only makes the problem easier), so that the observed label proportion $\widehat{\gamma}$ is simply $\gamma = \frac{1}{2}$. Then we seek the threshold $t'$ that minimizes

$$\text{EPR}(t) := \left| \mathbb{P}(f_t(X) = 1) - \frac{1}{2} \right|^p.$$

For any $p > 0$, $t'$ is the median of the marginal distribution of $X$, $\frac{1}{2}P_- + \frac{1}{2}P_+$, which equals $(\sqrt{5} - 1)/2 \approx 0.62 \neq t^*$. Thus, minimizing EPR does not yield an optimal classifier for BER or for misclassification rate, which agrees with BER in this setting where the two classes are equally likely.

Now suppose there are $N$ bags, with label proportions $\gamma_1, \ldots, \gamma_N$ drawn iid from a distribution whose (population) mean and median are $\frac{1}{2}$, such as the uniform distribution on $[0, 1]$. The optimal BER classifier remains the same, with threshold $t^* = \frac{1}{2}$. The optimal classifier wrt misclassification rate is also the same, assuming we view $\mathbb{E}[\gamma_i] = \frac{1}{2}$ as the class prior. In the infinite bag-size limit, EPR would seek the threshold $t'$ that minimizes

$$\text{EPR}_N(t) := \frac{1}{N} \sum_{i=1}^{N} |\mathbb{P}(f_t(X) = 1) - \gamma_i|^p.$$

For $p = 1$, EPR minimization selects $t'$ such that $\mathbb{P}(f_{t'}(X) = 1)$ is the empirical median of $\gamma_1, \ldots, \gamma_N$, which will be near $\frac{1}{2}$, which means $t'$ will be near 0.62. For $p = 2$, EPR minimization selects $t'$ such that $\mathbb{P}(f_{t'}(X) = 1)$ is the empirical mean of $\gamma_1, \ldots, \gamma_N$, which will again be near $\frac{1}{2}$, which again means $t'$ will be near 0.62.

More generally, based on the above example, EPR seems likely to fail whenever $P_+$ and $P_-$ are not sufficiently "symmetric."

## S2   Proofs of Results From Main Document

This section contains the proofs.

## S2.1 Proof of Proposition 1

Consider the loss function $\tilde{\ell}$ given by

$$\tilde{\ell}_+(t) = A\ell_+(t) - B\ell_-(t),$$
$$\tilde{\ell}_-(t) = C\ell_-(t) - D\ell_+(t).$$

Equating $\mathcal{E}_{P^\kappa}^{\tilde{\ell}}(f)$ to $\mathcal{E}_P^\ell(f)$ yields four equations in the four unknowns $A, B, C$, and $D$, corresponding to the coefficients of $\mathbb{E}_{X \sim P_\pm} \ell_\pm(f(X))$. The unique solution to this system is $\tilde{\ell} = \ell^\kappa$.

## S2.2 Proof of Proposition 4

We begin with $\mathcal{F}_{R,K}^k$. For any $R > 0$, $f \in \mathcal{F}_{R,K}^k$, and $x \in \mathcal{X}$,

$$|f(x)| = |\langle f, k(\cdot, x)\rangle| \le \|f\|_{\mathcal{H}} \|k(\cdot, x)\|_{\mathcal{H}} = RK.$$

by the reproducing property and Cauchy-Schwarz. Thus $A = RK$.

For the second part, the expectation may be bounded by a modification of the standard bound of Rademacher complexity for kernel classes. Thus,

$$\mathbb{E}_{(\epsilon_i)}\left[\sup_{f \in \mathcal{F}_{R,K}^k} \sum_i a_i \epsilon_i f(x_i)\right] = \mathbb{E}_{(\epsilon_i)}\left[\sup_{f \in \mathcal{F}_{R,K}^k} \sum_i a_i \epsilon_i \langle f, k(\cdot, x_i)\rangle\right] \tag{S1}$$

$$= \mathbb{E}_{(\epsilon_i)}\left[\sup_{f \in \mathcal{F}_{R,K}^k} \left\langle f, \sum_i a_i \epsilon_i k(\cdot, x_i)\right\rangle\right]$$

$$= \mathbb{E}_{(\epsilon_i)}\left[\left\langle R\frac{\sum_i a_i \epsilon_i k(\cdot, x_i)}{\|\sum_i a_i \epsilon_i k(\cdot, x_i)\|}, \sum_i a_i \epsilon_i k(\cdot, x_i)\right\rangle\right] \tag{S2}$$

$$= R\mathbb{E}_{(\epsilon_i)}\left[\sqrt{\left\|\sum_i a_i \epsilon_i k(\cdot, x_i)\right\|^2}\right]$$

$$\le R\sqrt{\mathbb{E}_{(\epsilon_i)}\left[\left\|\sum_i a_i \epsilon_i k(\cdot, x_i)\right\|^2\right]} \tag{S3}$$

$$= R\sqrt{\sum_i a_i^2 \|k(\cdot, x_i)\|^2} \tag{S4}$$

$$\le RK\sqrt{\sum_{i=1}^M a_i^2}, \tag{S5}$$

where (S1) uses the reproducing property, (S2) is the condition for equality in Cauchy-Schwarz, (S3) is Jensen's inequality, (S4) follows from independence of the Rademacher random variables, and (S5) follows from the reproducing property and the bound on the kernel.

Next, consider $\mathcal{F}^{\mathrm{NN}}_{\alpha,\beta}$. For the first part we have for any $f \in \mathcal{F}^{\mathrm{NN}}_{\alpha,\beta}$ and $x \in \mathcal{X}$,

$$
\begin{aligned}
|f(x)| &= |\langle v, [Ux]_+ \rangle| \\
&\leq \|v\| \|[Ux]_+\| \\
&\leq \|\alpha\| \|[Ux]_+\| \\
&\leq \|\alpha\| \|Ux\| \\
&= \|\alpha\| \sqrt{\sum_j |\langle u_j, x \rangle|^2} \\
&\leq \|\alpha\| \sqrt{\sum_j \|u_j\|^2 \|x\|^2} \\
&\leq \|\mathcal{X}\| \|\alpha\| \sqrt{\sum_j \|u_j\|^2} \\
&\leq \|\mathcal{X}\| \|\alpha\| \|\beta_j\|.
\end{aligned}
$$

For the second part, observe

$$
\begin{aligned}
\mathbb{E}_{(\epsilon_k)}\left[ \sup_{f \in \mathcal{F}} \sum_{k=1}^{M} \epsilon_k a_k f(x_k) \right] &= \mathbb{E}_{(\epsilon_k)}\left[ \sup_{f \in \mathcal{F}} \sum_{k=1}^{M} \epsilon_k a_k \sum_{j=1}^{h} v_j \left[ \langle u_j, x_k \rangle \right]_+ \right] \\
&= \mathbb{E}_{(\epsilon_k)}\left[ \sup_{f \in \mathcal{F}} \sum_{k=1}^{M} \epsilon_k \sum_{j=1}^{h} v_j \left[ \langle u_j, a_k x_k \rangle \right]_+ \right] \\
&= \mathbb{E}_{(\epsilon_k)}\left[ \sup_{f \in \mathcal{F}} \sum_{j=1}^{h} v_j \sum_{k=1}^{M} \epsilon_k \left[ \langle u_j, a_k x_k \rangle \right]_+ \right] \\
&\leq \mathbb{E}_{(\epsilon_k)}\left[ \sup_{f \in \mathcal{F}} \left| \sum_{j=1}^{h} v_j \sum_{k=1}^{M} \epsilon_k \left[ \langle u_j, a_k x_k \rangle \right]_+ \right| \right] \\
&\leq \mathbb{E}_{(\epsilon_k)}\left[ \sup_{f \in \mathcal{F}} \sum_{j=1}^{h} \alpha_j \left| \sum_{k=1}^{M} \epsilon_k \left[ \langle u_j, a_k x_k \rangle \right]_+ \right| \right] \\
&\leq \sum_{j=1}^{h} \alpha_j \mathbb{E}_{(\epsilon_k)} \sup_{f \in \mathcal{F}} \left| \sum_{k=1}^{M} \epsilon_k \left[ \langle u_j, a_k x_k \rangle \right]_+ \right|. \\
&= \sum_{j=1}^{h} \alpha_j \mathbb{E}_{(\epsilon_k)} \sup_{u_j : \|u_j\| \leq \beta_j} \left| \sum_{k=1}^{M} \epsilon_k \left[ \langle u_j, a_k x_k \rangle \right]_+ \right|. \quad\quad \text{(S6)}
\end{aligned}
$$

We bound the expectations in (S6) using Ledoux-Talagrand contraction [6, Theorem 4.12].

**Theorem 1** (Ledoux-Talagrand contraction). *Let $F : \mathbb{R}_+ \to \mathbb{R}_+$ be convex and increasing. Further let $\varphi_i$, $i \in [M]$ be 1-Lipschitz functions such that $\varphi(0) = 0$. Then, for any bounded subset $T \subset \mathbb{R}^M$,*

$$
\mathbb{E}_{(\epsilon_i)} F\left( \frac{1}{2} \sup_{t \in T} \left| \sum_{i=1}^{M} \epsilon_i \varphi_i(t_i) \right| \right) \leq \mathbb{E}_{(\epsilon_i)} F\left( \sup_{t \in T} \left| \sum_{i=1}^{M} \epsilon_i t_i \right| \right).
$$

To apply this result, for each $j$ notice that

$$
\mathbb{E}_{(\epsilon_k)} \sup_{u_j : \|u_j\| \leq \beta_j} \left| \sum_{k=1}^{M} \epsilon_k \left[ \langle u_j, a_k x_k \rangle \right]_+ \right| = \mathbb{E}_{(\epsilon_k)} \sup_{t \in T_j} \left| \sum_{k=1}^{M} \epsilon_k [t_k]_+ \right|
$$

where $t = (t_1, t_2, \ldots, t_M)^T$ and

$$
T_j = \left\{ t = (\langle u_j, a_1 x_1 \rangle, \langle u_j, a_2 x_2 \rangle, \ldots, \langle u_j, a_M x_M \rangle)^T \in \mathbb{R}^M : \|u_j\| \leq \beta_j \right\}
$$

which is clearly bounded. Now taking $F$ to be the identity and $\varphi_i = [\cdot]_+$, we have

$$\mathbb{E}_{(\epsilon_k)} \sup_{u_j : \|u_j\| \leq \beta_j} \left| \sum_{k=1}^{M} \epsilon_k \left[ \langle u_j, a_k x_k \rangle \right]_+ \right| \leq 2\mathbb{E}_{(\epsilon_k)} \sup_{u_j : \|u_j\| \leq \beta_j} \left| \sum_{k=1}^{M} \epsilon_k \langle u_j, a_k x_k \rangle \right|$$

$$= 2\mathbb{E}_{(\epsilon_k)} \sup_{u_j : \|u_j\| \leq \beta_j} \left| \left\langle u_j, \sum_{k=1}^{M} \epsilon_k a_k x_k \right\rangle \right|$$

$$= 2\mathbb{E}_{(\epsilon_k)} \left\langle \beta_j \frac{\sum_{k=1}^{M} \epsilon_k a_k x_k}{\| \sum_{k=1}^{M} \epsilon_k a_k x_k \|}, \sum_{k=1}^{M} \epsilon_k a_k x_k \right\rangle$$

$$= 2\beta_j \mathbb{E}_{(\epsilon_k)} \sqrt{\left\| \sum_{k=1}^{M} \epsilon_k a_k x_k \right\|^2} \tag{S7}$$

$$\leq 2\beta_j \sqrt{\mathbb{E}_{(\epsilon_k)} \left\| \sum_{k=1}^{M} \epsilon_k a_k x_k \right\|^2} \tag{S8}$$

$$\leq 2\beta_j \sqrt{\sum_{k=1}^{M} a_k^2 \|x_k\|^2} \tag{S9}$$

$$\leq 2\|\mathcal{X}\|_2 \beta_j \sqrt{\sum_{k=1}^{M} a_k^2}, \tag{S10}$$

where (S7) uses the condition for equality in Cauchy-Schartz, (S8) uses Jensen's inequality, and (S9) uses independence of the $\epsilon_k$. The result now follows from (S6) and (S10).

## S2.3   Proof of Theorem 5

We first review the following properties of the supremum which are easily verified.

P1  For any real-valued functions $f_1, f_2 : \mathcal{X} \to \mathbb{R}$,
$$\sup_x f_1(x) - \sup_x f_2(x) \leq \sup_x (f_1(x) - f_2(x)).$$

P2  For any real-valued functions $f_1, f_2 : \mathcal{X} \to \mathbb{R}$,
$$\sup_x (f_1(x) + f_2(x)) \leq \sup_x f_1(x) + \sup_x f_2(x).$$

P3  $\sup(\cdot)$ is a convex function, i.e., if $(x_\lambda)_{\lambda \in \Lambda}$ and $(x'_\lambda)_{\lambda \in \Lambda}$ are two sequences (where $\Lambda$ is possibly uncountable), then $\forall \alpha \in [0, 1]$,
$$\sup_{\lambda \in \Lambda} (\alpha x_\lambda + (1 - \alpha) x'_\lambda) \leq \alpha \sup_{\lambda \in \Lambda} x_\lambda + (1 - \alpha) \sup_{\lambda \in \Lambda} x'_\lambda.$$

Introduce the variable $S$ to denote all realizations $X_{ij}^\sigma, 1 \in [N], \sigma \in \{-, +\}, j \in [n_i^\sigma]$. We would like to bound

$$\xi(S) := \sup_{f \in \mathcal{F}} \left| \sum_{i=1}^{N} w_i \left( \frac{1}{2} \sum_{\sigma \in \{\pm 1\}} \left[ \frac{1}{n_i^\sigma} \sum_{j=1}^{n_i^\sigma} \ell_\sigma^\kappa(f(X_{ij}^\sigma)) \right] - \mathcal{E}(f) \right) \right|.$$

Introduce

$$\xi^+(S) := \sup_{f \in \mathcal{F}} \sum_{i=1}^{N} w_i \left( \frac{1}{2} \sum_{\sigma \in \{\pm 1\}} \left[ \frac{1}{n_i^\sigma} \sum_{j=1}^{n_i^\sigma} \ell_\sigma^\kappa(f(X_{ij}^\sigma)) \right] - \mathcal{E}(f) \right),$$

$$\xi^-(S) := \sup_{f \in \mathcal{F}} -\sum_{i=1}^{N} w_i \left( \frac{1}{2} \sum_{\sigma \in \{\pm 1\}} \left[ \frac{1}{n_i^\sigma} \sum_{j=1}^{n_i^\sigma} \ell_\sigma^\kappa(f(X_{ij}^\sigma)) \right] - \mathcal{E}(f) \right).$$

Assume **(IIM)** holds. Since the realizations $X_{ij}^\sigma$ are independent, we can apply the Azuma-McDiarmid bounded difference inequality [7] to $\xi^+$ and to $\xi^-$. We will show that the same bound on $\xi^+$ and $\xi^-$ holds with probability at least $1 - \delta/2$. Combining these bounds gives the desired bound on $\xi$. We consider $\xi^+$ below, with the analysis for $\xi^-$ being identical.

**Definition 2.** *Let $A$ be some set and $\phi : A^n \to R$. We say $\phi$ satisfies the bounded difference assumption if $\exists c_1, \ldots, c_n \geqslant 0$ s.t. $\forall i, 1 \leqslant i \leqslant n$*

$$\sup_{x_1,\ldots,x_n,x_i' \in A} |\phi(x_1,\ldots,x_i,\ldots,x_n) - \phi(x_1,\ldots,x_i',\ldots,x_n)| \leqslant c_i$$

*That is, if we substitute $x_i$ to $x_i'$, while keeping other $x_j$ fixed, $\phi$ changes by at most $c_i$.*

**Lemma 3** (Bounded Difference Inequality)**.** *Let $X_1, \ldots, X_n$ be arbitrary independent random variables on set $A$ and $\phi : A^n \to R$ satisfy the bounded difference assumption. Then $\forall t > 0$*

$$\Pr\{\phi(X_1,\ldots,X_n) - \mathbb{E}[\phi(X_1,\ldots,X_n)] \geqslant t\} \leqslant e^{-\frac{2t^2}{\sum_{i=1}^n c_i^2}}.$$

To apply this result to $\xi^+$, first note that for any $f \in \mathcal{F}, x \in \mathcal{X}$, and $y \in \{-1, 1\}$,

$$\begin{aligned}
|\ell^{\kappa_i}(f(x), y)| &\leq |\ell^{\kappa_i}(0, y)| + |\ell^{\kappa_i}(f(x), y) - \ell^{\kappa_i}(0, y)| \\
&\leq |\ell^{\kappa_i}|_0 + |\ell^{\kappa_i}||f(x)| \\
&\leq |\ell^{\kappa_i}|_0 + |\ell^{\kappa_i}|A.
\end{aligned}$$

If we modify $S$ by replacing some $X_{ij}^\sigma$ with another $X'$, while leaving all other values in $S$ fixed, then (by P1) $\xi^+$ changes by at most $2\frac{w_i(|\ell^{\kappa_i}|_0 + |\ell^{\kappa_i}|A)}{2n_i^\sigma}$, and we obtain that with probability at least $1 - \delta/2$ over the draw of $S_1, \ldots, S_N$,

$$\begin{aligned}
\xi^+ - \mathbb{E}\left[\xi^+\right] &\leq 2\sqrt{\frac{1}{2}\sum_{i=1}^N \frac{w_i^2(|\ell^{\kappa_i}|_0 + |\ell^{\kappa_i}|A)^2}{\bar{n}_i}\frac{\log(2/\delta)}{2}} \\
&\leq 2(1 + A|\ell|)\sqrt{\frac{1}{2}\sum_{i=1}^N \frac{w_i^2}{\bar{n}_i(1 - \kappa_i^- - \kappa_i^+)^2}\frac{\log(2/\delta)}{2}},
\end{aligned}$$

where we have used $|\ell^{\kappa_i}|_0 \leq 1/(1 - \kappa_i^- - \kappa_i^+)$ and $|\ell^{\kappa_i}| \leq |\ell|/(1 - \kappa_i^- - \kappa_i^+)$.

To bound $\mathbb{E}[\xi^+]$ we will use ideas from Rademacher complexity theory. Thus let $S'$ denote a separate (ghost) sample of corrupted data $(\underline{X}_{ij}^\sigma) \overset{iid}{\sim} \tilde{P}_\sigma^{\kappa_i}, i = 1, \ldots, N, \sigma \in \{\pm\}, j = 1, \ldots, n_i^\sigma$, independent of the realizations in $S$. Let $\widehat{\mathbb{E}}_S[f]$ be shorthand for $\sum_i w_i \sum_{\sigma \in \{\pm\}} \frac{1}{2n_i^\sigma} \sum_j \ell_\sigma^{\kappa_i}(f(X_{ij}^\sigma))$. Denote by $(\epsilon_{ij}^\sigma)\, i \in [N], \sigma \in \{\pm\}, j \in [n_i^\sigma]$, iid Rademacher variables (independent from everything else), and

let $\mathbb{E}_{(\epsilon_{ij}^\sigma)}$ denote the expectation with respect to all of these variables. We have

$$\mathbb{E}\left[\xi^+\right] = \mathbb{E}_S\left[\sup_{f\in\mathcal{F}}\sum_{i=1}^N w_i\left(\left[\sum_{\sigma\in\{\pm\}}\frac{1}{2n_i^\sigma}\sum_{j=1}^{n_i^\sigma}\ell_\sigma^{\kappa_i}(f(X_{ij}^\sigma))\right]-\mathcal{E}_P^\ell(f)\right)\right]$$

$$= \mathbb{E}_S\left[\sup_{f\in\mathcal{F}}\left(\widehat{\mathbb{E}}_S[f]-\mathbb{E}_{S'}\left[\widehat{\mathbb{E}}_{S'}[f]\right]\right)\right]$$

(by writing $\mathcal{E}_P^\ell(f)=\sum w_i\mathcal{E}_{P^{\kappa_i}}^{\ell^{\kappa_i}}(f)$ and applying Prop. 1 for each $i$)

$$\leq \mathbb{E}_{S,S'}\left[\sup_{f\in\mathcal{F}}\left(\widehat{\mathbb{E}}_S[f]-\widehat{\mathbb{E}}_{S'}[f]\right)\right]$$

(by P3 and Jensen's inequality)

$$= \mathbb{E}_{S,S'}\left[\sup_{f\in\mathcal{F}}\left(\sum_{i=1}^N w_i\sum_{\sigma\in\{\pm\}}\frac{1}{2n_i^\sigma}\sum_{j=1}^{n_i^\sigma}\ell_\sigma^{\kappa_i}(f(X_{ij}^\sigma))-\ell_\sigma^{\kappa_i}(f(\underline{X}_{ij}^\sigma))\right)\right]$$

$$= \mathbb{E}_{S,S',(\epsilon_{ij}^\sigma)}\left[\sup_{f\in\mathcal{F}}\left(\sum_{i=1}^N w_i\sum_{\sigma\in\{\pm\}}\frac{1}{2n_i^\sigma}\sum_{j=1}^{n_i^\sigma}\epsilon_{ij}^\sigma\left(\ell_\sigma^{\kappa_i}(f(X_{ij}^\sigma))-\ell_\sigma^{\kappa_i}(f(\underline{X}_{ij}^\sigma))\right)\right)\right]$$

(for all $i,\sigma,j$, $X_{ij}^\sigma$ and $\underline{X}_{ij}^\sigma$ are iid, and $\epsilon_{ij}^\sigma$ are symmetric)

$$\leq \mathbb{E}_{S,S',(\epsilon_{ij}^\sigma)}\left[\sup_{f\in\mathcal{F}}\sum_{i=1}^N w_i\sum_{\sigma\in\{\pm\}}\frac{1}{2n_i^\sigma}\sum_{j=1}^{n_i^\sigma}\epsilon_{ij}^\sigma\ell_\sigma^{\kappa_i}(f(X_{ij}^\sigma))\right]$$

$$+ \mathbb{E}_{S,S',(\epsilon_{ij}^\sigma)}\left[\sup_{f\in\mathcal{F}}\sum_{i=1}^N w_i\sum_{\sigma\in\{\pm\}}\frac{1}{2n_i^\sigma}\sum_{j=1}^{n_i^\sigma}(-\epsilon_{ij}^\sigma)\ell_\sigma^{\kappa_i}(f(\underline{X}_{ij}^\sigma))\right]$$

(by P2)

$$= 2\mathbb{E}_S\mathbb{E}_{(\epsilon_{ij}^\sigma)}\left[\sup_{f\in\mathcal{F}}\sum_{i=1}^N w_i\sum_{\sigma\in\{\pm\}}\frac{1}{2n_i^\sigma}\sum_{j=1}^{n_i^\sigma}\epsilon_{ij}^\sigma\ell_\sigma^{\kappa_i}(f(X_{ij}))\right].$$

To bound the innermost expectation we use the following result from Meir and Zhang [8].

**Lemma 4.** *Suppose* $\{\phi_t\},\{\psi_t\},t=1,\ldots,T$, *are two sets of functions on a set* $\Theta$ *such that for each* $t$ *and* $\theta,\theta'\in\Theta$, $|\phi_t(\theta)-\phi_t(\theta')|\leq|\psi_t(\theta)-\psi_t(\theta')|$. *Then for all functions* $c:\Theta\to\mathbb{R}$,

$$\mathbb{E}_{(\epsilon_t)}\left[\sup_\theta\left\{c(\theta)+\sum_{t=1}^T\epsilon_t\phi_t(\theta)\right\}\right]\leq\mathbb{E}_{(\epsilon_t)}\left[\sup_\theta\left\{c(\theta)+\sum_{t=1}^T\epsilon_t\psi_t(\theta)\right\}\right].$$

Switching from the single index $t$ to our three indices $i$, $\sigma$, and $j$, we apply the lemma with $\Theta=\mathcal{F}$, $\theta=f$, $c(\theta)=0$, $\phi_{ij}^\sigma(\theta)=\frac{w_i}{2n_i^\sigma}\ell_\sigma^{\kappa_i}(f(X_{ij}^\sigma))$, and $\psi_{ij}^\sigma(\theta)=\frac{w_i|\ell|}{2n_i^\sigma(1-\kappa_i^--\kappa_i^+)}f(X_{ij}^\sigma)$, where we use $|\ell_\sigma^{\kappa_i}|\leq|\ell|/(1-\kappa_i^--\kappa_i^+)$. This yields

$$\mathbb{E}\left[\xi^+\right]\leq 2\mathbb{E}_S\mathbb{E}_{(\epsilon_{ij}^\sigma)}\left[\sup_{f\in\mathcal{F}}\sum_{i=1}^N\frac{w_i|\ell|}{1-\kappa_i^--\kappa_i^+}\sum_{\sigma\in\{\pm\}}\frac{1}{2n_i^\sigma}\sum_{j=1}^{n_i^\sigma}\epsilon_{ij}^\sigma f(X_{ij}^\sigma)\right]$$

$$= 2\mathfrak{R}_c^I(\mathcal{F}),$$

To see the second inequality in (3), by **(SR)** we have

$$2\mathfrak{R}_c^I(\mathcal{F}) \leq 2B|\ell| \sqrt{\sum_{i,\sigma,j} \left( \frac{w_i}{2n_i^\sigma(1 - \kappa_i^- - \kappa_i^+)} \right)^2}$$

$$= 2B|\ell| \sqrt{\sum_i \frac{w_i^2}{4(1 - \kappa_i^- - \kappa_i^+)^2} \sum_\sigma \frac{1}{n_i^\sigma}}$$

$$= 2B|\ell| \sqrt{\sum_i \frac{w_i^2}{2\bar{n}_i(1 - \kappa_i^- - \kappa_i^+)^2}}$$

$$= \sqrt{2}B|\ell| \sqrt{\sum_i \frac{w_i^2}{\bar{n}_i(1 - \kappa_i^- - \kappa_i^+)^2}},$$

This concludes the proof in the **(IIM)** case.

Now assume **(IBM)** holds. The idea is to apply the bounded difference inequality at the MCM level. If we modify $S$ by replacing $X_{ij}^\sigma$ (with $i$ fixed, $j, \sigma$ variable) with other values $(X_{ij}^\sigma)'$, while leaving all other values in $S$ fixed, then (by P1) $\xi^+$ changes by at most $2w_i(|\ell^{\kappa_i}|_0 + |\ell^{\kappa_i}|A)$, and we obtain that with probability at least $1 - \delta/2$ over the draw of $S$,

$$\xi^+ - \mathbb{E}\left[\xi^+\right] \leq \sqrt{\sum_{i=1}^N w_i^2(|\ell^{\kappa_i}|_0 + |\ell^{\kappa_i}|A)^2 \frac{\log(2/\delta)}{2}}$$

$$\leq (1 + A|\ell|) \sqrt{\frac{\log(2/\delta)}{2}} \sqrt{\sum_{i=1}^N \frac{w_i^2}{(1 - \kappa_i^- - \kappa_i^+)^2}}.$$

To bound $\mathbb{E}\left[\xi^+\right]$, we use the same reasoning as in the **(IIM)** case to arrive at

$$\mathbb{E}\left[\xi^+\right] \leq 2\mathbb{E}_S \mathbb{E}_{(\epsilon_i)} \left[ \sup_{f \in \mathcal{F}} \sum_{i=1}^N w_i \epsilon_i \sum_{\sigma \in \{\pm\}} \frac{1}{2n_i^\sigma} \sum_{j=1}^{n_i^\sigma} \ell_\sigma^{\kappa_i}(f(X_{ij})) \right],$$

where now there is a Rademacher variable for every bag. The inner two summations may be expressed

$$\mathbb{E}_{(\sigma,X) \sim \widehat{P}^{\kappa_i}}\left[\ell_\sigma^{\kappa_i}(f(X))\right]$$

and so by Jensen's inequality and Lemma 4 we have

$$\mathbb{E}\left[\xi^+\right] \leq 2\mathbb{E}_S \mathbb{E}_{(\epsilon_i)} \left[ \sup_{f \in \mathcal{F}} \sum_{i=1}^N w_i \mathbb{E}_{(\sigma,X) \sim \widehat{P}^{\kappa_i}}\left[\ell_\sigma^{\kappa_i}(f(X))\right] \right]$$

$$\leq 2\mathbb{E}_S \mathbb{E}_{((\sigma_i, X_i) \sim \widehat{P}^{\kappa_i})_{i \in [N]}} \mathbb{E}_{(\epsilon_i)} \left[ \sup_{f \in \mathcal{F}} \sum_{i=1}^N \epsilon_i w_i \ell_{\sigma_i}^{\kappa_i}(f(X_i)) \right]$$

$$\leq 2\mathbb{E}_S \mathbb{E}_{((\sigma_i, X_i) \sim \widehat{P}^{\kappa_i})_{i \in [N]}} \mathbb{E}_{(\epsilon_i)} \left[ \sup_{f \in \mathcal{F}} \sum_{i=1}^N \epsilon_i \frac{w_i|\ell|}{1 - \kappa_i^- - \kappa_i^+} f(X_i) \right]$$

$$= 2\mathfrak{R}_c^B(\mathcal{F})$$

This proves the first inequality. To prove the second, by **(SR)** we have

$$2\mathfrak{R}_c^B(\mathcal{F}) \leq 2B|\ell| \sqrt{\sum_i \frac{w_i^2}{(1 - \kappa_i^- - \kappa_i^+)^2}}.$$

This concludes the proof.

## S2.4 Proof of Theorem 6

We begin by stating a generalization of Chernoff's bound to correlated binary random variables [11, 5].

**Lemma 5.** *Let $Z_1, \ldots, Z_m$ be binary random variables. Suppose there exists $0 \le \tau \le 1$ such that for all $I \subset [m]$, $\mathbb{P}(\prod_{i \in I} Z_i = 1) \le \tau^{|I|}$. Then for any $\epsilon \ge 0$, $\mathbb{P}(\sum_{i=1}^m Z_i \ge m(\tau + \epsilon)) \le e^{-2m\epsilon^2}$.*

We will first prove the theorem for BP. The result for dominating schemes will then follow easily. Thus, assume the $K$-merging scheme is BP. For now assume **(CIBM)** , which is implied by **(CIIM)** .

Let $\widehat{\gamma}_{ik}^+$ be the larger of the two *empirical* label proportions within the $k$th pair of small bags within the $i$th pair of big bags, and similarly let $\widehat{\gamma}_{ik}^-$ be the smaller. Also let $\gamma_{ik}^+$ be the larger of the two *true* label proportions within the $k$th pair of small bags within the $i$th pair of big bags, and similarly let $\gamma_{ik}^-$ be the smaller.

Let $\epsilon_0 \in (0, \Delta(1-\tau))$ and let $\epsilon \in (0, \frac{\Delta(1-\tau)-\epsilon_0}{1+\Delta}]$. For $i \in [M]$, let $K_i$ be the number of original pairs in the $i$th block (the $i$th pair of big bags) for which $|\gamma_{ik}^+ - \gamma_{ik}^-| \ge \Delta$, $k \in [K]$ and define $\Omega_{\gamma,i}$ to be the event that $K_i \ge K(1 - \tau - \epsilon)$. By Lemma 5 and **(LP)** , we have $\Pr_\gamma(\Omega_{\gamma,i}^c) \le e^{-2K\epsilon^2}$.

Also define $\Omega_{Y,i}$ to be the event that $\widehat{\Gamma}_i^+ - \widehat{\Gamma}_i^- \ge \mathbb{E}_{Y|\gamma}[\widehat{\Gamma}_i^+ - \widehat{\Gamma}_i^-] - \epsilon = \Gamma_i^+ - \Gamma_i^- - \epsilon$. Note that conditioned on $\gamma$, $\widehat{\Gamma}_i^+ - \widehat{\Gamma}_i^- = \frac{1}{K} \sum_{k=1}^K (\widehat{\gamma}_{ik}^+ - \widehat{\gamma}_{ik}^-)$ is the sum of $K$ independent random variables with range $[0, 1]$ (here we use the definition of BP and conditional independence of the small bags under **(CIBM)** ). By Hoeffding's inequality, $\mathbb{P}_{Y|\gamma}(\Omega_{Y,i}^c) \le e^{-2K\epsilon^2}$.

Now define $\Omega_\gamma := \bigcap_{i=1}^M \Omega_{\gamma,i}$ and $\Omega_Y := \bigcap_{i=1}^M \Omega_{Y,i}$. Also define $\Theta$ to be the event that the first inequality in (4) does not hold. Then

$$
\begin{aligned}
\mathbb{P}(\Theta) &\le \mathbb{P}(\Theta | \Omega_\gamma \cap \Omega_Y) + \mathbb{P}((\Omega_\gamma \cap \Omega_Y)^c) \\
&\le \mathbb{P}(\Theta | \Omega_\gamma \cap \Omega_Y) + \mathbb{P}(\Omega_\gamma^c) + \mathbb{P}(\Omega_Y^c) \\
&\le \mathbb{P}(\Theta | \Omega_\gamma \cap \Omega_Y) + \frac{N}{K} e^{-2K\epsilon^2} + \mathbb{E}_\gamma \mathbb{E}_{Y|\gamma} \left[ \mathbf{1}_{\{\Omega_Y^c\}} \right] \\
&\le \mathbb{P}(\Theta | \Omega_\gamma \cap \Omega_Y) + \frac{2N}{K} e^{-2K\epsilon^2} \\
&= \mathbb{E}_{\gamma,Y} \left[ \mathbb{E}_{X|\gamma,Y} \left[ \mathbf{1}_{\{\Theta\}} | \gamma, Y \right] | \Omega_\gamma \cap \Omega_Y \right] + \frac{2N}{K} e^{-2K\epsilon^2}.
\end{aligned}
$$

We next bound the inner expectation of the last line above, which is the conditional probability of $\Theta$ given fixed values of $(\gamma, Y) \in \Omega_\gamma \cap \Omega_Y$. We will bound this probability the same argument as in the proof of Thm. 5. To apply that argument, we first need to confirm two things: Conditioned on $\gamma, Y$, (1) for each $i$, $\widehat{\Gamma}_i^+ - \widehat{\Gamma}_i^- > 0$, and (2) the empirical error $\tilde{\mathcal{E}}(f)$ is an unbiased estimate of $\mathcal{E}_P^\ell$. The first property is given by the following.

**Lemma 6.** *Conditioned on $(\gamma, Y) \in \Omega_\gamma \cap \Omega_Y$, for all $i \in [M]$*

$$
\widehat{\Gamma}_i^+ - \widehat{\Gamma}_i^- \ge \Gamma_i^+ - \Gamma_i^- - \epsilon \ge \epsilon_0.
$$

*Proof.* Fix $(\gamma, Y) \in \Omega_\gamma \cap \Omega_Y$. Let $i \in [M]$. By definition of $\Omega_Y$,

$$
\begin{aligned}
\widehat{\Gamma}_i^+ - \widehat{\Gamma}_i^- &\ge \mathbb{E}_{Y|\gamma}[\widehat{\Gamma}_i^+ - \widehat{\Gamma}_i^-] - \epsilon \\
&= \left( \frac{1}{K} \sum_{k=1}^K \mathbb{E}_{Y|\gamma}[\widehat{\gamma}_{ik}^+ - \widehat{\gamma}_{ik}^-] \right) - \epsilon \\
&\ge \left( \frac{1}{K} \sum_{k=1}^K \gamma_{ik}^+ - \gamma_{ik}^- \right) - \epsilon.
\end{aligned}
$$

To see the last step, let $U$ and $V$ be random variables with means $p$ and $q$. Then $\mathbb{E}[\max(U, V) - \min(U, V)] = \mathbb{E}[|U - V|] \ge |\mathbb{E}[U - V]| = |p - q| = \max(p, q) - \min(p, q)$, by Jensen's inequality. Here we have again used the definitions of BP and **(CIBM)** .

By definition of $\Omega_\gamma$, $\gamma_{ik}^+ - \gamma_{ik}^- \geq \Delta$ for $K_i \geq K(1-\tau-\epsilon)$ values of $k \in [K]$. From this we conclude that $\widehat{\Gamma}_i^+ - \widehat{\Gamma}_i^- \geq \Delta(1-\tau-\epsilon) - \epsilon \geq \epsilon_0$, where the last step follows from $\epsilon \leq \frac{\Delta(1-\tau)-\epsilon_0}{1+\Delta}$. $\qquad\square$

For the second property, recall $\tilde{\mathcal{E}}(f) = \sum_i w_i \tilde{\mathcal{E}}_i(f)$ with $w \in \Delta^M$ and $w_i \propto (\widehat{\Gamma}_i^+ - \widehat{\Gamma}_i^-)^2$. We note that $\mathbb{E}_{\boldsymbol{X}|\boldsymbol{\gamma},\boldsymbol{Y}\in\Omega_\gamma\cap\Omega_{\boldsymbol{Y}}}\left[\tilde{\mathcal{E}}_i(f)\right]$ is well defined because $|\ell^{\widehat{\kappa}_i}(f(x))|$ is bounded for $x \in \mathcal{X}$. This follows from the assumption $\sup_{f\in\mathcal{F},x\in\mathcal{X}}|f(x)| \leq A < \infty$, the fact that $\ell^{\widehat{\kappa}_i}$ is Lipschitz continuous on $\Omega_\gamma \cap \Omega_{\boldsymbol{Y}}$ by Lemma 6, and the observation $|\ell^{\widehat{\kappa}_i}(f(x))| \leq |\ell^{\widehat{\kappa}_i}|_0 + |\ell^{\widehat{\kappa}_i}|A$.

**Lemma 7.** *For all $f \in \mathcal{F}$, $\mathbb{E}_{\boldsymbol{X}|\boldsymbol{\gamma},\boldsymbol{Y}\in\Omega_\gamma\cap\Omega_{\boldsymbol{Y}}}\left[\tilde{\mathcal{E}}_i(f)\right] = \mathcal{E}_P^\ell(f)$.*

*Proof.* Recall that $X_{mj}$ denotes the $j$th instance in the $m$th original (pre-merging) small bag, $m \in [2N]$, $j \in [n]$, and that $Y_{mj}$ denotes the corresponding label. We have

$$\mathbb{E}_{\boldsymbol{X}|\boldsymbol{\gamma},\boldsymbol{Y}\in\Omega_\gamma\cap\Omega_{\boldsymbol{Y}}}\left[\tilde{\mathcal{E}}_i(f)\right]$$

$$= \frac{1}{2}\mathbb{E}_{\boldsymbol{X}|\boldsymbol{\gamma},\boldsymbol{Y}\in\Omega_\gamma\cap\Omega_{\boldsymbol{Y}}}\left[\frac{1}{nK}\sum_{m\in I_i^+}\sum_{j=1}^n \ell_+^{\widehat{\kappa}_i}(f(X_{mj})) + \frac{1}{nK}\sum_{m\in I_i^-}\sum_{j=1}^n \ell_-^{\widehat{\kappa}_i}(f(X_{mj}))\right]$$

$$= \frac{1}{2}\mathbb{E}_{\boldsymbol{X}|\boldsymbol{\gamma},\boldsymbol{Y}\in\Omega_\gamma\cap\Omega_{\boldsymbol{Y}}}\left[\widehat{\Gamma}_i^+\frac{1}{nK\widehat{\Gamma}_i^+}\sum_{m\in I_i^+}\sum_{j:Y_{mj}=1}\ell_+^{\widehat{\kappa}_i}(f(X_{mj}))\right.$$

$$+ (1-\widehat{\Gamma}_i^+)\frac{1}{nK(1-\widehat{\Gamma}_i^+)}\sum_{m\in I_i^+}\sum_{j:Y_{mj}=-1}\ell_+^{\widehat{\kappa}_i}(f(X_{mj}))$$

$$+ \widehat{\Gamma}_i^-\frac{1}{nK\widehat{\Gamma}_i^-}\sum_{m\in I_i^-}\sum_{j:Y_{mj}=1}\ell_-^{\widehat{\kappa}_i}(f(X_{mj}))$$

$$\left.+ (1-\widehat{\Gamma}_i^-)\frac{1}{nK(1-\widehat{\Gamma}_i^-)}\sum_{m\in I_i^-}\sum_{Y_{mj}=-1}\ell_-^{\widehat{\kappa}_i}(f(X_{mj}))\right]$$

$$= \frac{1}{2}\left\{\widehat{\Gamma}_i^+\mathbb{E}_{X\sim P_+}\left[\ell_+^{\widehat{\kappa}_i}(f(X))\right] + (1-\widehat{\Gamma}_i^+)\mathbb{E}_{X\sim P_-}\left[\ell_+^{\widehat{\kappa}_i}(f(X))\right]\right.$$

$$\left.+ \widehat{\Gamma}_i^-\mathbb{E}_{X\sim P_+}\left[\ell_-^{\widehat{\kappa}_i}(f(X))\right] + (1-\widehat{\Gamma}_i^-)\mathbb{E}_{X\sim P_-}\left[\ell_-^{\widehat{\kappa}_i}(f(X))\right]\right\}$$

$$= \frac{1}{2}\left\{\mathbb{E}_{X\sim P_+^{\widehat{\kappa}_i}}\left[\ell_+^{\widehat{\kappa}_i}(f(X))\right] + \mathbb{E}_{X\sim P_-^{\widehat{\kappa}_i}}\left[\ell_-^{\widehat{\kappa}_i}(f(X))\right]\right\}$$

$$= \mathcal{E}_P^\ell(f)$$

where the third step uses the definition of **(CIBM)** , and the last step uses Prop. 1 and Lemma 6. $\quad\square$

By Lemmas 6 and Lemma 7, we can apply the argument in the proof of Theorem 5, conditioned on $(\boldsymbol{\gamma},\boldsymbol{Y}) \in \Omega_\gamma \cap \Omega_{\boldsymbol{Y}}$, with the estimator $\tilde{\mathcal{E}}$ instead of $\widehat{\mathcal{E}}_w$. The only other changes are that in the application of Lemma 4, we use the bound

$$|\ell^{\widehat{\kappa}_i}| \leq \frac{|\ell|}{\widehat{\Gamma}_i^+ - \widehat{\Gamma}_i^-} \leq \frac{|\ell|}{\Gamma_i^+ - \Gamma_i^- - \epsilon},$$

and in the final bounds, we upper bound $(\widehat{\Gamma}_i^+ - \widehat{\Gamma}_i^-)^{-1}$ by $(\Gamma_i^+ - \Gamma_i^- - \epsilon)^{-1}$.

## S3  Symmetric Losses

A loss is said to by *symmetric* if there exists a constant $K$ such that for all $t$, $\ell(t,1) + \ell(t,-1) = K$. Examples include the 0-1, sigmoid, and ramp losses. For a symmetric loss, $\ell^\kappa$ simplifies to

$$\ell^\kappa(t,y) = \frac{1}{1-\kappa^+-\kappa^-}\ell(t,y) - \frac{K}{1-\kappa^+-\kappa^-}(\kappa^-\mathbf{1}_{\{y=1\}} + \kappa^+\mathbf{1}_{\{y=-1\}}).$$

Combined with Proposition 1, this yields

$$\mathcal{E}_{P^\kappa}^\ell(f) = (1 - \kappa^+ - \kappa^-)\mathcal{E}_P^\ell(f) + K\left(\frac{\kappa^+ + \kappa^-}{2}\right).$$

Therefore, the two sides have the same minimizer which implies that the BER is *immune* to label noise under a mutual contamination model. That is, training on the contaminated data without modifying the loss still minimizes the clean BER. This result has been previously observed for the 0/1 loss [9] and general symmetric losses [14, 2]. The above argument gives a simple derivation from Prop. 1.

## S4  Convexity

We say that the loss $\ell$ is *convex* if, for each $\sigma$, $\ell_\sigma(t)$ is a convex function of $t$. Let $\ell_\sigma''$ denote the second derivative of $\ell$ with respect to its first variable. The condition in (S11) below was used by Natarajan et al. [10] to prove a convexity result an unbiased loss in the class-conditional noise setting. Here we prove a version for MCMs.

**Proposition 8.** *Suppose $\kappa^- + \kappa^+ < 1$ and let $\ell$ be a convex, twice differentiable loss satisfying*

$$\ell_+''(t) = \ell_-''(t). \tag{S11}$$

*If $\kappa^\sigma < \frac{1}{2}$ for $\sigma \in \{\pm\}$, then $\ell^\kappa$ is convex.*

Examples of losses satisfying the second order condition include the logistic, Huber, and squared error losses. The result is proved by simply observing

$$(\ell_\sigma^\kappa)''(t) = \ell_+''(t)\frac{1 - 2\kappa^{-\sigma}}{1 - \kappa^- - \kappa^+}$$
$$\geq 0.$$

The statement about $\widehat{\mathcal{E}}_i(f)$ being convex when $f$ is linear was a holdover from an earlier draft and should be disregarded. In the infinite bag size limit, $\widehat{\mathcal{E}}_i(f)$ converges to $\mathcal{E}_P^\ell(f)$, which is convex in the output of $f$ provided $\ell$ is convex. Sufficient conditions for the convexity of $\widehat{\mathcal{E}}_i(f)$ or $\widehat{\mathcal{E}}_w(f)$ for small bag sizes is an interesting open question.

## S5  (CIBM') implies (IBM)

Assume that **(CIBM')** holds. To show **(IBM)**, we need to show that for a fixed bag $i$, and for all $j \in [n_i]$, the marginal distribution of $X_{ij}$, conditioned on the bag, is $\gamma_i P_+ + (1 - \gamma_i)P_-$. Thus let $A$ be an arbitrary event. Also let $p_i$ be the joint pmf of $Y_{i1}, \ldots, Y_{in_i}$, conditioned on the bag. Without loss of generality let $j = 1$. We have

$$
\begin{aligned}
\mathbb{P}(X_{i1} \in A) &= \mathbb{E}_X\big[\mathbf{1}_{\{X_{i1} \in A\}}\big] \\
&= \mathbb{E}_{Y_{i1},\ldots,Y_{in_i}} \mathbb{E}_{X_{i1}|Y_{i1},\ldots,Y_{in_i}}\big[\mathbf{1}_{\{X_{i1} \in A\}}\big] \\
&= \mathbb{E}_{Y_{i1},\ldots,Y_{in_i}} \mathbb{P}_{Y_{i1}}(X_{i1} \in A) \tag{S12} \\
&= \sum_{(y_1,\ldots,y_{n_i})\in\{-1,1\}^{n_i}} \mathbb{P}_{y_1}(X_{i1} \in A)p_i(y_1,\ldots,y_{n_i}) \\
&= P_+(A) \sum_{(y_2,\ldots,y_{n_i})\in\{-1,1\}^{n_i-1}} p_i(1, y_2, \ldots, y_{n_i}) \\
&\quad + P_-(A) \sum_{(y_2,\ldots,y_{n_i})\in\{-1,1\}^{n_i-1}} p_i(-1, y_2, \ldots, y_{n_i}) \\
&= \gamma_i P_+(A) + (1 - \gamma_i)P_-(A), \tag{S13}
\end{aligned}
$$

where (S12) and (S13) use **(CIMB')**.

## S6 Optimal Bag Matching

The bound is minimized by selecting weights

$$w_i \propto \bar{n}_i (\gamma_i^+ - \gamma_i^-)^2,$$

which gives preference to pairs of bags where one bag is mostly +1's (large $\gamma_i^+$) and the other is mostly -1's (small $\gamma_i^-$). With these weights, the **(SR)** bound is proportional to under **(CIIM)**

$$\sqrt{\left( \sum_{i=1}^{N} \bar{n}_i (\gamma_i^+ - \gamma_i^-)^2 \right)^{-1}}.$$

Here and below, under **(CIBM')'** substitute $\bar{n}_i \to 1$.

We can optimize the pairing of bags by further optimizing the bound. Consider the unpaired bags $(B_i, \gamma_i)$, $i = 1, \ldots, 2N$. Recall that $\bar{n}_i = \mathrm{HM}(n_i^+, n_i^-)$. We would like to pair each bag to a different bag, forming pairs $(\gamma_i^+, \gamma_i^-)$, such that

$$\sum_{i=1}^{N} \bar{n}_i (\gamma_i^+ - \gamma_i^-)^2$$

is maximized. For each $i < j$, let $u_{ij}$ be a binary variable, with $u_{ij} = 1$ indicating that the $i$th and $j$th bags are paired. The optimal pairing of bags is given by the solution to the following integer program:

$$\max_{u} \sum_{1 \le i < 2N} \sum_{i < j \le 2N} \mathrm{HM}(n_i, n_j)(\gamma_i - \gamma_j)^2 u_{ij} \tag{S14}$$

$$\text{s.t. } u_{ij} \in \{0, 1\}, \forall i, j$$

$$\sum_{i<j} u_{ij} + \sum_{j<i} u_{ji} = 1, \forall i.$$

The equality constraint ensures that every bag is paired with precisely one other distinct bag. This problem is known as the "maximum weighted (perfect) matching" problem. An exact algorithm to solve it was given by Edmonds [4], and several approximate algorithms also exist for large scale problems [3].

When $n_i^\sigma = n$ for all $i$ and $\sigma$, the solution to this integer program is very simple.

**Proposition 9.** *If $n_i^\sigma = n$ for all $i$ and $\sigma$, then the solution to* (S14) *is to match the largest $\gamma_i$ with the smallest, the second largest $\gamma_i$ with the second smallest, and so on.*

*Proof.* Suppose the statement is false. Then there exists an optimal solution, and $i$ and $j$, such that $\gamma_i^+ > \gamma_j^+$ and $\gamma_i^- > \gamma_j^-$. Now consider the matching obtained by swapping the bags associated to $\gamma_i^-$ and $\gamma_j^-$. Then the objective function increases by

$$(\gamma_i^+ - \gamma_j^-)^2 + (\gamma_j^+ - \gamma_i^-)^2 - (\gamma_i^+ - \gamma_i^-)^2 - (\gamma_j^+ - \gamma_j^-)^2 = 2(\gamma_i^+ - \gamma_j^+)(\gamma_i^- - \gamma_j^-) > 0.$$

This contradicts the assumed optimality. □

## S7 Merging Schemes that Dominate Blockwise-Pairwise

Let $\underline{\Gamma}_i^+$ and $\underline{\Gamma}_i^-$ denote the quantities $\Gamma_i^+$ and $\Gamma_i^-$ when the merging scheme is BP, and let $\Gamma_i^+$ and $\Gamma_i^-$ refer to any other merging scheme under consideration. Similarly, let $\underline{\widehat{\Gamma}}_i^+$ and $\underline{\widehat{\Gamma}}_i^-$ denote the quantities $\widehat{\Gamma}_i^+$ and $\widehat{\Gamma}_i^-$ when the merging scheme is BP, and let $\widehat{\Gamma}_i^+$ and $\widehat{\Gamma}_i^-$ refer to any other merging scheme under consideration.

For a $K$-merging scheme that dominates BP, we still have $\widehat{\Gamma}_i^+ - \widehat{\Gamma}_i^- \ge \underline{\Gamma}_i^+ - \underline{\Gamma}_i^- - \epsilon \ge \epsilon_0 > 0$ on $\Omega_\gamma \cap \Omega_Y$ by definition of dominating. Hence the same proof goes through in this case, and we may state the following.

**Theorem 10.** *Let* (**LP**) *hold. Let* $\epsilon_0 \in (0, \Delta(1 - \tau))$. *Let* $\ell$ *be a Lipschitz loss and let* $\mathcal{F}$ *satisfy* $\sup_{x \in \mathcal{X}, f \in \mathcal{F}} |f(x)| \leq A < \infty$. *Let* $\epsilon \in (0, \frac{\Delta(1-\tau)-\epsilon_0}{1+\Delta}]$ *and* $\delta \in (0, 1]$. *For any* $K$-*merging scheme that dominates* $BP$, *under* (**CIIM**), *with probability at least* $1 - \delta - 2\frac{N}{K}e^{-2K\epsilon^2}$ *with respect to the draw of* $\gamma, \boldsymbol{Y}, \boldsymbol{X}$,

$$\widehat{\Gamma}_i^+ - \widehat{\Gamma}_i^- \geq \underline{\Gamma}_i^+ - \underline{\Gamma}_i^- - \epsilon \geq \epsilon_0$$

*and*

$$\sup_{f \in \mathcal{F}} \left| \tilde{\mathcal{E}}(f) - \mathcal{E}(f) \right| \leq 2\mathfrak{R}_c^I(\mathcal{F}) + C\sqrt{\frac{\mathrm{HM}((\underline{\Gamma}_i^+ - \underline{\Gamma}_i^- - \epsilon)^{-2})}{(N/K)n}} \overset{(\mathbf{SR})}{\leq} D\sqrt{\frac{\mathrm{HM}((\underline{\Gamma}_i^+ - \underline{\Gamma}_i^- - \epsilon)^{-2})}{(N/K)n}}, \tag{S15}$$

*where* $c_i = w_i|\ell|/(\underline{\Gamma}_i^+ - \underline{\Gamma}_i^- - \epsilon)$, $C = (1 + A|\ell|)\sqrt{\log(2/\delta)}$, *and* $D = 2B|\ell| + C$. *Under* (**CIBM**), *the same bounds hold with the same probability if we substitute* $\mathfrak{R}_c^I(\mathcal{F}) \rightarrow \mathfrak{R}_c^B(\mathcal{F})$ *and* $n \rightarrow 1$.

We conjecture that it is possible to improve the bound for dominating schemes. Using the current proof technique, this would require proving that

$$\widehat{\Gamma}_i^+ - \widehat{\Gamma}_i^- \geq \Gamma_i^+ - \Gamma_i^- - \epsilon$$

with high probability. For example, with BM, this would require a one-sided tail inequality for how the difference between the average of the larger half and the average of the smaller half of $2K$ independent random variables deviates from its mean. The BP scheme was selected as a reference because it is straightforward to prove such a bound for BP using Hoeffding's inequality.

## S8 Consistency

A discrimination rule $\widehat{f}$ is (weakly) consistent if $\mathcal{E}_P^\ell(\widehat{f}) \rightarrow \inf_f \mathcal{E}_P^\ell(f)$ in probability as $N \rightarrow \infty$, where the infimum is over all decision functions.

We first note that if we desire consistency wrt the BER defined with 0-1 loss, it suffices to prove consistency wrt the BER defined with a loss $\ell$ that is "classification calibrated" [1], such as the logistic loss. This is because the BER corresponds to a special case of the usual misclassification risk when the class probabilities are equal.

We state our consistency result for the discrimination rule

$$\widehat{f} \in \arg\min_{f \in \mathcal{F}} J(f) := \tilde{\mathcal{E}}(f) + \lambda\|f\|_{\mathcal{F}_k}^2,$$

where $\mathcal{F}_k$ is the reproducing kernel Hilbert space associated to a symmetric, positive definite kernel, and $\lambda > 0$.

**Theorem 11.** *Let* $\mathcal{X}$ *be compact and let* $k$ *be a bounded, universal kernel on* $\mathcal{X}$. *Let* $K \rightarrow \infty$ *such that* $N/K \rightarrow \infty$ *and* $N = O(K^\beta)$ *for some* $\beta > 0$, *as* $N \rightarrow \infty$. *Let* $\lambda$ *be such that* $\lambda \rightarrow 0$ *and* $\lambda(N/K)/\log(N/K) \rightarrow \infty$ *as* $N \rightarrow \infty$. *Let* (**LP**) *and* (**CIBM**) *hold. Then for any merging scheme that dominates* $BP$,

$$\mathcal{E}(\widehat{f}) \rightarrow \inf_f \mathcal{E}_P^\ell(f) \tag{S16}$$

*in probability as* $N \rightarrow \infty$.

*Proof.* Let $B$ denote the bound on the kernel. By Proposition 4 and by Theorem 10 applied to $\mathcal{F}_{B,R}^k$, for all $\epsilon_0 \in (0, \Delta(1 - \tau))$, $\epsilon \in (0, \frac{\Delta(1-\tau)-\epsilon_0}{1+\Delta}]$, and $\delta \in (0, 1]$, with probability at least $1 - \delta - \frac{N}{K}e^{-2K\epsilon^2}$,

$$\sup_{f \in B_k(R)} \left| \tilde{\mathcal{E}}(f) - \mathcal{E}_P^\ell(f) \right| \leq \frac{D}{\epsilon_0}\sqrt{\frac{K}{N}}$$

where $D = (1 + RB|\ell|)\sqrt{\log(2/\delta)} + 2RB|\ell|$.

Observe that $J(\widehat{f}) \leq J(0) \leq \frac{|\ell|_0}{\epsilon_0}$. Therefore $\lambda\|\widehat{f}\|^2 \leq \frac{|\ell|_0}{\epsilon_0} - \tilde{\mathcal{E}}(\widehat{f}) \leq \frac{2|\ell|_0}{\epsilon_0}$ and so $\|\widehat{f}\|^2 \leq \frac{2|\ell|_0}{\epsilon_0\lambda}$.

Set $R = \sqrt{\frac{2|\ell|_0}{\epsilon_0\lambda}}$. Note that $R$ grows asymptotically because $\lambda$ shrinks. We just saw that $\widehat{f} \in B_k(R)$.

Let $\epsilon > 0$. Fix $f_\epsilon \in \mathcal{F}_k$ s.t. $\mathcal{E}_P^\ell(f_\epsilon) \leq \inf_f \mathcal{E}_P^\ell + \epsilon/2$, possible since $k$ is universal [13]. Note that $f_\epsilon \in B_k(R)$ for $N$ sufficiently large. In this case the generalization error bound implies that with probability $\geq 1 - \delta - \frac{N}{K}e^{-2K\epsilon^2}$,

$$\mathcal{E}_P^\ell(\widehat{f}) \leq \tilde{\mathcal{E}}(\widehat{f}) + \frac{D}{\epsilon_0}\sqrt{\frac{K}{N}}$$

$$\leq \tilde{\mathcal{E}}(f_\epsilon) + \lambda\|f_\epsilon\|^2 - \lambda\|\widehat{f}\|^2 + \frac{D}{\epsilon_0}\sqrt{\frac{K}{N}}$$

$$\leq \tilde{\mathcal{E}}(f_\epsilon) + \lambda\|f_\epsilon\|^2 + \frac{D}{\epsilon_0}\sqrt{\frac{K}{N}}$$

$$\leq \mathcal{E}_P^\ell(f_\epsilon) + \lambda\|f_\epsilon\|^2 + \frac{2D}{\epsilon_0}\sqrt{\frac{K}{N}}.$$

Taking $\delta = K/N$, the result now follows. $\qquad\qquad\qquad\qquad\qquad\qquad\qquad\square$

## S9  Experimental Details

The parameters of InvCal [12] and alter-$\propto$SVM [15] are tuned by five-fold cross validation. We only consider the RBF kernel. Following [15], the parameters for both methods were set as follows. The kernel bandwidth $\gamma$ of the RBF kernel is chosen from $\{0.01, 0.1, 1\}$. For InvCal, the parameters are tuned from $C_p \in \{0.1, 1, 10\}$, and $\epsilon \in \{0, 0.01, 0.1\}$. For alter-$\propto$SVM, the parameters are tuned from $C \in \{0.1, 1, 10\}$, and $C_p \in \{1, 10, 100\}$.

A Matlab implementation of both InvCal and alter-$\propto$SVM was obtained online.[1] These implementations rely on LIBSVM[2] and CVX[3]. We modified the code to preform parameter tuning with cross validation as described above. The modified code is contained in our supplemental material. LIBSVM contains its own random number generator that was unfortunately not seeded and hence the results for alter-$\propto$SVM are not reproducible.

All three algorithms require random initialization. Yu et al. [15] randomly initialize their algorithm ten times and take the result with smallest objective value. This would take over 10 hours on 100 cores to run in out setup. Hence, we only consider one random initialization for each method. This could account for the relatively poor performance of alter-$\propto$SVM.

We also found that in some cases, the code for alter-$\propto$-SVM wouldn't create a variable 'support_v', which is used to predict the test label. The resulted from LIBSVM not returning any support vectors. If 'support_v' did not exist for a given fold, we excluded that fold from the cross-validation error estimate.

For bag size 8, in the experiments reported below, on a handful of occasions there are only two bags in the validation data within a given fold of cross-validation, and both bags have the same label proportion. When this occurs, we cannot compute our criterion, and exclude such folds.

For the MAGIC dataset, InvCal takes roughly 30 minutes on 36 cores to complete the experiments for all bag sizes. For the Adult dataset, InvCal takes roughly 60 minutes on 36 cores. For alter-$\propto$-SVM, the approximated runtime on MAGIC dataset is 70 minutes on 144 cores. On Adult dataset, it is 100 minutes on 144 cores.

## S10  Additional Experimental Results

We performed an additional set of experiments where the number of bags $N$ remains fixed. For Adult dataset, the total number of bags is 16, and for MAGIC, it is 12. For each method, we generate an ROC curve and evaluate the area under the curve (AUC) using the test data. The average AUCs and the standard deviations over 5 random trials are reported in Table S1. Bold numbers indicate that a method's mean AUC was the largest for that experimental setting. We observe that LMMCM exhibits excellent performance in this setting as well.

Table S1: AUC. Column header indicates bag size.

| Data set, LP dist | Method | 8 | 32 | 128 | 512 |
|---|---|---|---|---|---|
| Adult, $\left[0, \frac{1}{2}\right]$ | InvCal | $0.6427 \pm 0.0922$ | $0.6545 \pm 0.0643$ | $0.6518 \pm 0.0139$ | $0.7230 \pm 0.0253$ |
| | alter-$\propto$SVM | $0.6525 \pm 0.0817$ | $0.5959 \pm 0.1145$ | $0.6199 \pm 0.1267$ | $0.6419 \pm 0.0997$ |
| | LMMCM | $\mathbf{0.7299 \pm 0.0796}$ | $\mathbf{0.7765 \pm 0.0590}$ | $\mathbf{0.8329 \pm 0.0166}$ | $\mathbf{0.8456 \pm 0.0213}$ |
| Adult, $\left[\frac{1}{2}, 1\right]$ | InvCal | $0.5973 \pm 0.0740$ | $0.6634 \pm 0.0864$ | $0.6408 \pm 0.0216$ | $0.7218 \pm 0.0170$ |
| | alter-$\propto$SVM | $0.6035 \pm 0.1626$ | $\mathbf{0.7774 \pm 0.0443}$ | $0.5863 \pm 0.2775$ | $0.7106 \pm 0.2193$ |
| | LMMCM | $\mathbf{0.7228 \pm 0.1048}$ | $0.7674 \pm 0.0586$ | $\mathbf{0.8428 \pm 0.0101}$ | $\mathbf{0.8588 \pm 0.0091}$ |
| MAGIC, $\left[0, \frac{1}{2}\right]$ | InvCal | $\mathbf{0.7381 \pm 0.0439}$ | $0.7828 \pm 0.0212$ | $0.7936 \pm 0.0371$ | $0.8196 \pm 0.0231$ |
| | alter-$\propto$SVM | $0.5997 \pm 0.1163$ | $0.5376 \pm 0.1671$ | $0.6859 \pm 0.0371$ | $0.7193 \pm 0.1278$ |
| | LMMCM | $0.7180 \pm 0.0450$ | $\mathbf{0.7852 \pm 0.7828}$ | $\mathbf{0.8140 \pm 0.0463}$ | $\mathbf{0.8630 \pm 0.0275}$ |
| MAGIC, $\left[\frac{1}{2}, 1\right]$ | InvCal | $0.6741 \pm 0.0673$ | $0.7405 \pm 0.0433$ | $0.7876 \pm 0.0249$ | $0.8135 \pm 0.0132$ |
| | alter-$\propto$SVM | $0.6589 \pm 0.1029$ | $0.6330 \pm 0.1254$ | $0.6790 \pm 0.1072$ | $0.7965 \pm 0.0708$ |
| | LMMCM | $\mathbf{0.6807 \pm 0.0779}$ | $\mathbf{0.7639 \pm 0.0335}$ | $\mathbf{0.7905 \pm 0.0258}$ | $\mathbf{0.8491 \pm 0.0245}$ |

## Footnotes

[1] https://github.com/felixyu/pSVM

[2] https://www.csie.ntu.edu.tw/ cjlin/libsvm/

[3] http://cvxr.com/cvx/