[Reviews · NeurIPS 2020]

Review 1

Summary and Contributions: The authors study the problem of learning from label proportions by using mutual contamination models. To this end, they propose a new unbiased loss and are able to derive generalization bounds under non-iid samples. A weighted empirical risk then arises naturally that is consistent for LLP. Experiments are simulated where there is an important improvement in the scenario of larger bag sizes.

Strengths: 1. The theoretical contributions are sound. I checked some of the proofs in more or less detail and could not spot any errors. 2. The idea of mixing MCM with LLP is a novel contribution and authors derived generalization bounds and proposed a consistent learning procedure. 3. The content is mostly theoretical with some experiments that support the theory. Thus, the work is relevant to the learning theory community and can also be of interest to practitioners.

Weaknesses: * There are no comments on the computational complexity of their method and how it compares to previous work. * There are no comments on the tightness or optimality of the generalization bound.

Correctness: Yes, theoretical and empirical results are sound.

Clarity: Yes, the paper is well written. I enjoyed reading it.

Relation to Prior Work: Authors do a good job in highlighting the difference between each of their contributions and existing work.

Reproducibility: Yes

Additional Feedback: 1. Given that their method uses bag-pairing, k-merging schemes. I would like to see some discussion on the computational complexity and how it compares to previous methods. For instance, can this method be scaled? (My guess is that it could be difficult given the use of ILP) 2. Optimality of bounds can be another paper on its own, but do the authors have some comments on it? 3. I might be missing something here, but in Line 292 I read that LLPs are uniform on [0, 0.5] and on [0.5, 1]? Does this mean that in the experiments we do not have, say, a label proportion of 0.2 and another label proportion of 0.8? If this is the case, why? 4. Would be worth looking at the precision-recall AUC in those cases? I am curious what would be the performances. === UPDATE After rebuttal, I am keeping my score unchanged. It's not clear if the results would hold when using approximate solvers for ILP in the case of unequal sized bags. Thus, I still have a concern on the practical usability of the method *with* the given guarantees and cannot vote for a greater score.


Review 2

Summary and Contributions: The paper encapsulates learning from label proportions in a more general framework, called mutual contamination framework. It is shown that if one knows the contamination rates (akin to the proportions in the LLP framework), then the models can be learned to minize the EBR loss over instances. Concentration inequalities in the fashion of Rademacher complexity are then given to show that efficient training can take place in practice, with the bounds suggesting ways to make optimal choices in the method parameters. Experiments show that the method performs quite well, in particular with large-sized bags.

Strengths: * An encompassing framework for LLP, allowing to derive consistency results under quite general conditions. * Corresponding learning algorithms that show a quite good performances.

Weaknesses: * Not much, the paper is well-written, with a first part accessible to a wide audience, and a second part focusing on technical results of the paper. As said by the authors, one could think of a multi-class setting, yet the binary case is an essential and important first step towards it.

Correctness: * I am not an expert in Rademacher complexity, but the results appeared correct to me.

Clarity: * Although the second part gets quite technical, the paper is overall well-written, with clear explanations about the interest and limitations of the results.

Relation to Prior Work: * Yes

Reproducibility: Yes

Additional Feedback: * Unfortunately for the authors, this is a bit far from my expertise to give meaningful comments (I could not check this before, due to the COVID situation).


Review 3

Summary and Contributions: The paper deals with a weakly supervised learning problem referred to as Learning from Label Proportions (LLP), akin to standard multiclass classification except that data are grouped into datasets (‘bags’) and only the class proportions are at disposal for learning. The approach developed in the article is that of Mutual Contamination Models (MCMs). It is described in section 2 in the binary case: data within a bag being generated from a certain mixture of the two class distributions, bags being independent from each other. Two situations are considered, depending on whether the data within a bag are independent or not. The goal pursued is to minimise the balanced error rate (BER), i.e. the error rate when the parameter mixture is equal ti 1/2. Generalisation bounds are obtained in terms of BER and involve specific complexity measures described in Definition 3. In 3.1, it is explained that, when the true proportions are known that LLP boils down to applying results for MCMs by pairing bags, whereas the situation where only empirical proportions are available is considered in 3.2: a maximal deviation bound is stated in Theorem 6. Experimental results are displayed in section 4.

Strengths: The LLP framework is well motivated, the results seem to be novel and the parallel drawn with MCMs may be original.

Weaknesses: Regarding the form, the paper could be improved by being less dense. Although none of the arguments is complex, the paper is very hard to follow. A very general framework is introduced (cf dependence structure), whereas restrictive assumptions are next made (cf conditional independence): it is very difficult to know which hypotheses cannot be avoided. In addition, the form of the generalisation bounds is weird, because not formulated in terms of excess of risk, as if nothing could be said on the optimal learning strategy in the LLP framework. And the choice of BER seems arbitrary as well, insofar as a ROC analysis of the decision functions output by the algorithm proposed is carried out in the experimental section, The BER being just the AUC of the binary classifier, corresponding to a single point of the ROC curve of the decision function (where the tangent has slope equal to one), there is no reason for the whole ROC curve to be accurate everywhere.

Correctness: The paper seems correct to me but I am skeptical of the relevance of the BER analysis carried out here.

Clarity: The paper is understandable but could be improved (style and general organisation).

Relation to Prior Work: The state of the art is well described.

Reproducibility: Yes

Additional Feedback:


Review 4

Summary and Contributions: This paper provides a novel solver for LLP problem based on unbiased surrogate loss for balanced error rate (BER). The main contribution is on theorical side. I have checked the proofs in a general way, and I don’t find any obvious mistake. I think it will be better if the authors can summarize their contributions in the introduction part clearly, it will help the readers to catch the main results of this paper.

Strengths: although the authors claim that “the theoretical underpinnings of LLP have been slow to develop”, and they offer “a general-purpose, theoretically grounded empirical objective for training LLP classifiers”, it seems that they only extend the former mutual contamination model to the multiple models scenario, to seamlessly apply the results to solve LLP problems. I cannot find any novel insights on LLP specifically. As a result, I think this paper is to offer a new algorithm for LLP with better theorical grounds, compared with the existing LLP solvers.

Weaknesses: 1. The work is strongly based on the results for mutual contamination models, which is specially designed and discussed for binary classification problems. As a result, the benchmark approaches involved in the experimental section are confined in two early proposed methods for binary problems. Many up-to-date models that focus on multi-class LLP problems are not touched by this work. In other words, there is a gap between this work and multi-class LLP problem. In particular, thanks to deep neural networks, the performance on LLP has been greatly improved by recent work, such as the work in [1], [11], [18], [31], and [34], on much complicated datasets, e.g., image data. This is an obvious limitation of this work. The authors should notice this shortcoming and continue to provide an explanation or fix this problem in their future work. 2. Even for the binary case, the experiments are insufficient in this paper, and the results is not significantly better than that of the two algorithms, given they are out-of-date in solving LLP. I suggest adding more experimental results. For example, more experimental results on hyper-parameter analysis. In addition, for the K-merging schemes, I do not quite understand how it work and how the number K is fixed in the experiments. Please offer more insights or analysis on this issue for improvement.

Correctness: Yes

Clarity: Yes

Relation to Prior Work: Yes

Reproducibility: Yes

Additional Feedback:

[Author Response · NeurIPS 2020]

We thank the authors for their careful reading of the paper. Below we repeat or paraphrase the reviewers' comments and questions and then offer our responses.

**R1: Computational complexity, especially in light of integer linear programming.** Bag pairing does indeed reduce to ILP. In the case of bags of equal size (the setting of our experiments), we show in the supplemental that the optimal algorithm is extremely simple: Pair the bags with highest and lowest LPs, the bags with next highest and next lowest, etc. For unequal bag sizes, this no longer holds, but there is a literature on scalable approximate algorithms for the weighted matching problem. We'll reference this in the final version.

**R1: Tightness of bounds.** While we have not studied this issue in depth, we anticipate that there are certain scenarios (e.g., worst case ones) where our bounds are tight, similar to conventional Rademacher complexity. In more typical scenarios, we expect that the bounds could be improved, e.g., by an appropriate analogue of local Rademacher complexity. However, we chose Rademacher complexity because we found that it led to a tractable upper bound that we were able to explicitly optimize to determine the weights for the different bag pairs.

**R1: Distribution of LPs:** This is a design choice in setting up the experiment. The user specifies the distribution of LPs, and we chose ours to be uniform on the given intervals.

**R4: Restrictive assumptions (cf conditional independence):** Under both our models (IIM and IBM), we allow the distribution of an instance to be dependent on which bag it is drawn from. Furthermore, under IBM, the instances within a bag (conditioned on the bag label proportion) can have arbitrary dependency structure. The literature on LLP frequently asserts the importance of these two settings, and our paper is the first to provide theoretical analysis wrt a classification performance measure under these assumptions. We have made every effort to make the assumptions as general as possible, and the assumptions needed for our results are indicated in our theorem statements.

**R4: Generalisation bounds not formulated in terms of excess of risk.** The term "generalization error bound" commonly refers to a bound on the deviation between true and empirical values of a performance measure. Given such a bound, there are standard arguments that yield consistency wrt the performance measure of interest, in our case BER. The reviewer may be asking about a "calibration" type excess risk bound. Such a bound would relate the excess risk for BER with loss $\ell$, to excess risk for BER with 0-1 loss. If $\ell$ is calibrated wrt 0-1 loss (which is true for common losses like logistic), such bounds can easily be obtained for BER (see [5]). This aspect of our work is very similar to the analysis for the misclassification rate, and hence has been placed in the supplemental, although we will highlight it more prominently in the revision.

**R4: Why focus on BER when experiments optimize AUC?** The BER is needed for our theoretical analysis. The BER pairs naturally with MCMs, just like the misclassification rate pairs with the label flipping model for label noise [20]. BER is basically the frequentist analogue of misclassification rate, which assumes the class labels are governed by a prior distribution. The experiments use AUC because our competitors are designed wrt misclassification rate. We wanted a performance measure that did not favor one method over the other, so we chose to look at something other than BER or misclassification rate, and AUC seemed a natural choice.

**R5: Extensions to multiclass / deep learning.** We completely agree this is the next step. We wanted to present a thorough theoretical treatment and found it necessary to first understand the binary case. We have been working on the multiclass case and can confirm that while it is definitely a separate paper, many of the ideas from the binary case do extend with some interesting caveats.

**R5: Competitors out of date.** The two competitors are indeed somewhat old by ML standards, but they are, to our knowledge, the top-performing kernel methods that have appeared in mainstream machine learning publications. We will certainly develop a neural network implementation and compare to more recent methods in our future work on multiclass LLP.

**R5: $K$-Merging schemes.** Our experiment don't use $K$-merging schemes. We use Alg. 1, which is mentioned in line 221. How do $K$-merging schemes work? We would need to know more about your question. If you update your review with more details, we can address it in the next version of our paper.

[Meta-Review · NeurIPS 2020]

This paper provides learning theoretic guarantees for a semi-supervised or transfer-type problem of learning from "label proportions" where several bags of data are provided but without labels for every point, instead having access to the label proportion within each bag. The problem is clearly challenging to analyze theoretically, so the authors have taken on a difficult task and done a decent job. Three reviewers were warmly receptive (a fourth has low confidence), but nobody strongly arguing for rejection nor acceptance in the discussion. Overall, it is a nice paper, and regardless of the way this paper finally goes, the authors would benefit from discussing some of the finer details of their assumptions and theorems.